# Signal requirement for cortical potential of transplantable human neuroepithelial stem cells

Balazs V. Varga [1,2✉], Maryam Faiz[1,3], Helena Pivonkova [2,9], Gabriel Khelifi [4,5,9], Huijuan Yang[1], Shangbang Gao [1], Emma Linderoth[1], Mei Zhen [1], Ragnhildur Thora Karadottir[2,6], Samer M. Hussein[1,4,5] & Andras Nagy [1,7,8✉]

The cerebral cortex develops from dorsal forebrain neuroepithelial progenitor cells. Following the initial expansion of the progenitor cell pool, these cells generate neurons of all the cortical layers and then astrocytes and oligodendrocytes. Yet, the regulatory pathways that control the expansion and maintenance of the progenitor cell pool are currently unknown. Here we define six basic pathway components that regulate proliferation of cortically specified human neuroepithelial stem cells (cNESCs) in vitro without the loss of cerebral cortex developmental potential. We show that activation of FGF and inhibition of BMP and ACTIVIN A signalling are required for long-term cNESC proliferation. We also demonstrate that cNESCs preserve dorsal telencephalon-specific potential when GSK3, AKT and nuclear CATENIN-β1 activity are low. Remarkably, regulation of these six pathway components supports the clonal expansion of cNESCs. Moreover, cNESCs differentiate into lower- and upper-layer cortical neurons in vitro and in vivo. The identification of mechanisms that drive the neuroepithelial stem cell self-renewal and differentiation and preserve this potential in vitro is key to developing regenerative and cell-based therapeutic approaches to treat neurological conditions.

[1] Lunenfeld-Tanenbaum Research Institute, Mount Sinai Hospital, Toronto, ON, Canada. [2] Wellcome – MRC Cambridge Stem Cell Institute, University of Cambridge, Puddicombe Way, Cambridge, UK. [3] Department of Surgery, Institute of Medical Science, University of Toronto, Toronto, ON, Canada. [4] Cancer Research Center, Université Laval, Quebec City, QC, Canada. [5] CHU of Québec-Université Laval Research Center, Oncology Division, Quebec City, QC, Canada. [6] Department of Veterinary Medicine, University of Cambridge, Cambridge, UK. [7] Department of Obstetrics and Gynaecology, and Institute of Medical Science, University of Toronto, Toronto, ON, Canada. [8] Australian Regenerative Medicine Institute, Monash University, Melbourne, VIC, Australia. [9] These authors contributed equally: Helena Pivonkova, Gabriel Khelifi. ✉email: bv243@cam.ac.uk; nagy@lunenfeld.ca

The cerebral cortex develops from the forebrain vesicle of the mammalian neuroepithelium. Forebrain specification of the anterior neuroepithelium happens shortly after the neural commitment of the epiblast[1]. After this initial specification, dorsal forebrain neuroepithelial cells divide symmetrically to expand the progenitor cell pool[2,3]. Upon neural tube closure, cells divide both symmetrically to expand the progenitor cell pool and asymmetrically to produce differentiated cells. During early forebrain development, differentiating neural progenitors initially give rise to glutamatergic neurons of the six cortical layers, each with specific connectivity to other brain regions[4]. At later stages, progenitor cells stop producing neurons and start generating astrocytes and oligodendrocytes[5].

While the dynamics of neural cell fate production from dorsal forebrain neuroepithelial cells are well established, the molecular mechanisms that instruct their expansion and maintenance remain poorly understood. It is known that preservation of multipotent neural progenitor cells in the forebrain requires fibroblast growth factor (FGF) receptor activity[6]. However, neuroepithelial cells isolated from the embryonic rodent telencephalon show limited proliferation in FGF alone[7]. Complementation of FGF with undefined components of brain-vessel endothelial cell conditioned media can extend the proliferation capacity of mouse neuroepithelial progenitors and preserve their early cortical potential transiently, indicating that the cells can self-renew if the precise signals are present[8].

The investigation of neural progenitor cell potency, cortical development or pathogenesis in humans is restricted by ethical and technical boundaries. This limitation has been overcome by the in vitro use of human pluripotent stem cells; that can be differentiated to a neuroectodermal fate[9] to provide a source of various neural cell types[10]. Both monolayer and organoid-based methods were improved to make cultures containing progenitors, cortical neurons and glial cells[11–16] but the self-renewal of cortical progenitors was not accomplished or the differentiation potential of the proliferating progenitors was not cortical[17–19].

Here we identify six signalling pathway components that instruct permanent self-renewal and fate preservation of dorsal forebrain neuroepithelial stem cells. In addition to FGF, self-renewal requires the simultaneous inhibition of BMP and ACTIVIN A signalling, reduced GSK3 and AKT activity and low level of β-catenin (CTNNB1) nuclear activity. We achieve this complex regulation with six factors (6F condition). This results in permanent culture and expansion of cortically specified neuroepithelial cells (cNESCs), which express the dorsal telencephalic markers *FOXG1, PAX6, EMX2, LHX2* and *OTX1/2*, and give rise to glutamatergic projection neurons of lower and upper cortical layers, as well as oligodendrocytes and astrocytes. The 6F condition is sufficient for single cell-derived cNESC colony formation and preserves the embryonic dorsal forebrain-specific differentiation potential of the cells. We also demonstrate that developmental signalling pathways implicated in neural tissue development, such as EGF, SHH and NOTCH signalling, are not required for the self-renewal of the cNESCs. Transplantation of cNESCs into the mouse brain demonstrates that these cells can differentiate into active glutamatergic projection neurons in the adolescent cortex and receive synaptic inputs. Our results show that cells with dorsal forebrain specification can be maintained by selective pathway regulation for long-term in vitro and preserve their early developmental potential.

## Results

### Generation of cortical neuroepithelial cells.
We established dorsal forebrain-specific neuroepithelial stem cells from human pluripotent stem cells (hPSCs) using chemical inhibitors of TGFβ (TGFβRi) and BMP (BMPRi) signalling (Fig. 1a and Supplementary Fig. 1a–c, Supplementary Table 1). Similar to other neural induction protocols[9,20], efficient neural conversion of hPSCs was seen by day 8 in the majority of cells (>95%; Supplementary Fig. 1a), as indicated by the upregulation of neuroepithelial marker, SOX1 (Supplementary Fig. 1b,c), downregulation of pluripotency marker, OCT4 (Supplementary Fig. 1a), and maintenance of neural stem cell marker, SOX2. We confirmed neural specification by examining the relative levels of *OCT4* and *NESTIN*, a neural progenitor cell marker in day 11 differentiated NESCs compared to undifferentiated hPSCs. We observed decreased *OCT4* and increased *NESTIN* mRNA expression in induced NESCs (Supplementary Fig. 1d), which was similar to mRNA expression levels in human foetal forebrain-derived neural stem cells [CB660[21]]. When we examined forebrain specification in these cells, we found that induced NESCs showed increased expression of *EMX2, PAX6, FOXG1* and *OTX2*, region-specific transcription factor mRNAs that together mark the dorsal forebrain vesicle (Supplementary Fig. 1d), and no change in the expression of ventral forebrain markers *LHX6* and *NKX2.1*, ventral midbrain markers *FOXA2* and *EN1*, and hindbrain marker *HOXB1* (Supplementary Fig. 1d). Immunocytochemistry confirmed the expression of SOX1, SOX2, FOXG1, NESTIN and PAX6 markers (Fig. 1b–e). These results indicate efficient induction of dorsal forebrain/cortex specified NESCs from hPSCs within 8–11 days, which we termed cNESCs.

### Regulation of TGFβR and GSK3 regulates maintenance of cNESCs.
Several studies have shown that dorsal forebrain specification is lost during long-term culture of neural stem cells in FGF and EGF[17,21–23]. Therefore, we asked whether additional extracellular signals are required to maintain FGF-dependent self-renewal of cNESCs. We investigated modulation of the WNT, BMP/TGFβ, EGF, SHH, NOTCH pathways, which have all been implicated in the regulation of cortical excitatory neuron specification[24].

WNT signals are important positive regulators of forebrain fate[25] and induce cell proliferation. WNT signal is mediated via the inhibition of GSK3, which can upregulate CTNNB1 and BMP/TGFβ/SMAD signalling[26,27] and alter neuroepithelial cell specification. Therefore, we tested if FGF in combination with inhibitors of GSK3 (GSK3i), BMPRi and/or TGFβRi, could maintain the SOX1-positive cNESC population. After neural induction, cNESCs-derived from 3 different human Embryonic Stem Cell (hESC) lines (H1, H9[28] and CA1[29]) were propagated for five passages in neural maintenance media with FGF and various combination of inhibitors (Fig. 1f). Both FGF alone and FGF and EGF were insufficient to maintain a high ratio of SOX1 positive cells. Addition of (4) BMPRi, (5) TGFβRi, (6) GSK3i, (7) GSK3i and TGFβRi and (8) BMPRi and TGFβRi with FGF reduced the ratio of SOX1 positive cells, while addition of (9) FGF, GSK3i and BMPRi showed no change compared to cNESCs in (2) FGF alone or (3) FGF and EGF. Strikingly, we found that neural cells derived from all three hESC lines cultured in FGF and all three inhibitors (termed 4F for 4 factor condition) contained the highest proportion of SOX1-positive cNESCs (mean value 86%; Fig. 1f). Conditions 2, 4, 5, 6 and 8 did not support the propagation of at least one of the three different hPSC lines and led to significant cell death, suggesting cell line-dependent variation.

In agreement with other studies[17,21–23], we found that cNESCs showed limited proliferation in FGF alone. To determine whether 4F could improve the long-term proliferation of SOX1-positive cells we compared the growth of cNESCs in: (a) FGF alone, (b) EGF and FGF, (c) 4F using chemical inhibitors (FGF, LDN193189 (BMPRi), SB431542 (TGFβRi), CHIR99021 (GSK3i), abbreviated as FLSC) or d) 4F using protein antagonists (FGF,

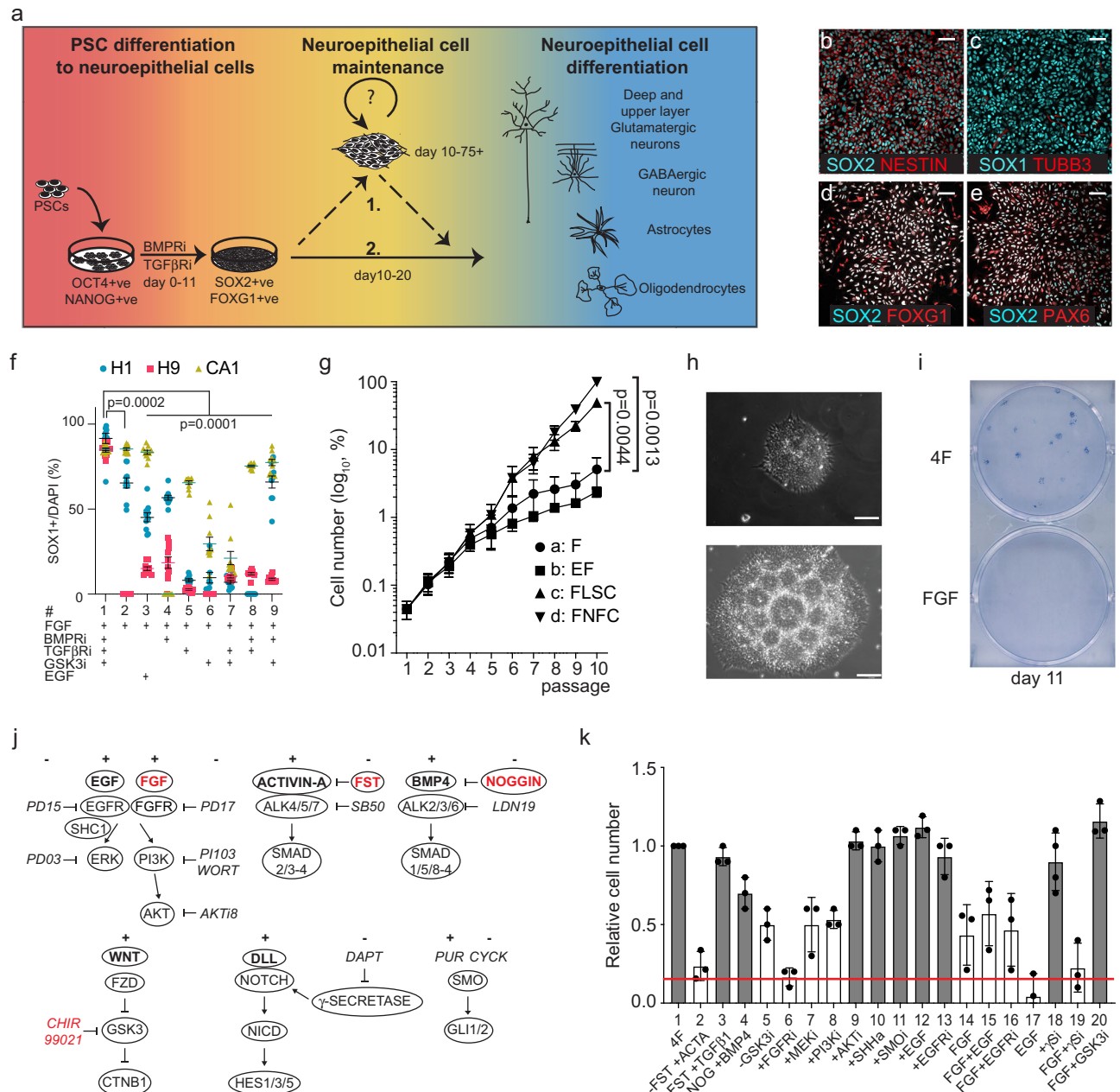

**Fig. 1 Induction and maintenance of SOX1 positive dorsal forebrain neuroepithelial cells. a** Neural induction scheme of hPSCs in the presence of TGFbR and BMPR inhibitors. After cortical neural induction NES cells express NESTIN in SOX2 positive cells **b** SOX1 in TUBB3 negative cells **c**, FOXG1 **d**, and PAX6 **e** in SOX2 positive cells. The experiment was repeated with 5 biologically independent cell lines. Scalebar: 25 μm **f** SOX1 positive cell ratio in passage 5 cultures of hES (H1, H9, CA1) derived cNESCs treated with various combination of factors ($n = 3$ independent cell lines, 10 datapoints per each cell line per group, red bars are mean ± SEM, one-way ANOVA, Tukey's test). **g** Normalised model of cell number changes of cNESCs (Data are presented as mean ± SEM of H1-, H9-, CA1-derived cells, $n = 3$ independent cell lines, 2-way ANOVA, Dunnett's test) over 10 passages (30 days). **h** Phase contrast image of H1 derived cNESCs (p36) in 4F on day 4 (top) and on day 14 of culture (bottom) forming rosette structures. The experiment was repeated with 4 biologically independent cell lines. Scalebar: 50 μm. **i:** Colony formation assay of cNESCs cultured in 4F prior to seeding at 200 cells/cm$^2$ density. **j** Schematic presentation of developmental signalling pathway components targeted in our assay, protein ligands are in bold and chemical inhibitors are in italic, 4F components are in red. **k** Quantification of cell number changes after 96-h treatment of cNESCs with indicated ligands or chemical inhibitors compared to 4F condition ($n = 3$ independent experiments, data are presented as mean ± SEM, 1 way ANOVA, Tukey's test, white bars are $p < 0.01$, grey bars are not significantly different from control, red line indicates starting cell number). Source data are provided as a Source Data file.

NOGGIN (BMPi), FOLLISTATIN (ACTIVINi), and CHIR99021 (GSK3i), abbreviated as FNFC. Confirming previous findings, after six passages, cNESCs in FGF alone and FGF and EGF had a lower proliferation rate than in previous passages. In contrast, cNESCs in both FLSC and FNFC showed sustained proliferation and supported the formation of neuroepithelial colonies (Fig. 1g, h). To confirm that the expansion of cNESCs in 4F was not associated with any specific chromosomal abnormalities we examined the karyotype of cNESCs maintained for over 100 days in vitro (more than 50 population doublings) in two independent cell lines and found no chromosomal aberrations (Supplementary Fig. 1e).

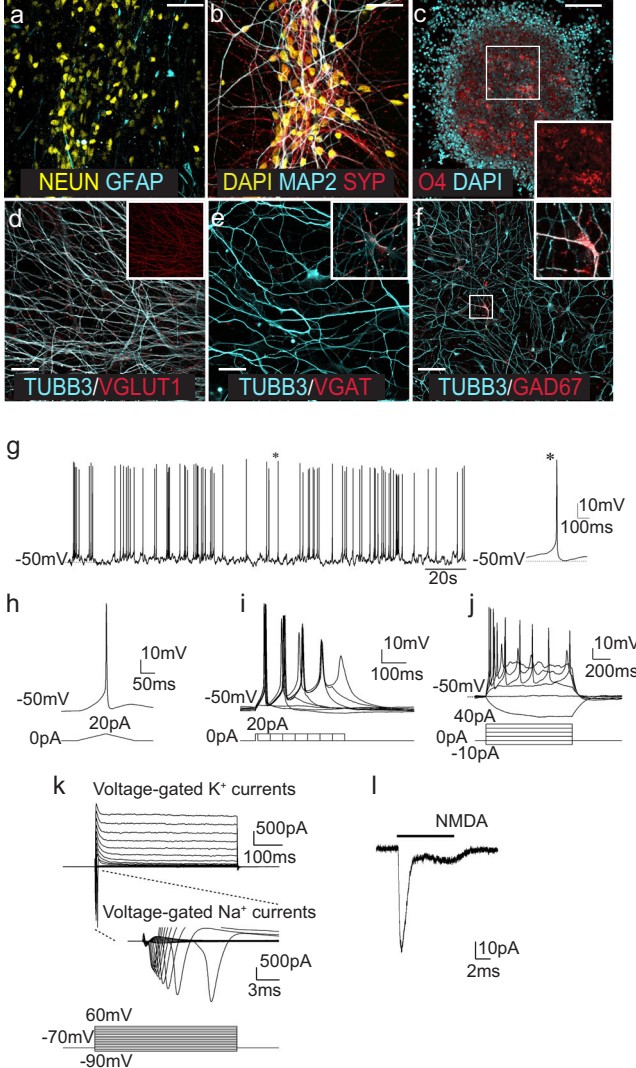

**Fig. 2 cNESCs are multipotent. a** cNESCs (1.53E) differentiate to NEUN positive neurons and GFAP positive astrocytes. The experiment was repeated with 3 biologically independent cell lines. Scalebar: 20 μm. **b** Mature MAP2 positive neurons express Synaptophysin (SYP) by day 30. The experiment was repeated with 3 biologically independent cell lines. Scalebar: 20 μm. **c** cNESCs (1.53E) differentiate to O4 positive oligodendrocytes, insert shows colony in higher magnification. The experiment was repeated with 2 biologically independent cell lines. Scalebar: 50 μm. **d** The majority of cNESC (H1A) neurons are glutamatergic projection neurons positive for vesicular glutamate transporter type 1 (VGLUT1), insert shows VGLUT1 labelling. The experiment was repeated with 4 biologically independent cell lines. Scalebar: 10 μm. **e** Among the projection neurons (negative for VGAT) a subpopulation of GABAergic interneurons can be found, expressing vesicular GABA transporter (VGAT, high magnification from a different area in the insert). The experiment was repeated with 3 biologically independent cell lines. Scalebar: 10 μm. **f:** GABAergic interneurons also express glutamate amino decarboxylase 67 enzyme (GAD67, the squared area is shown in high magnification in the insert). The experiment was repeated with 3 biologically independent cell lines. Scalebar: 20 μm. **g** Representative spontaneous action potentials (APs) in differentiated H1A cNESC neurons. A single typical action potential was shown in the right panel (indicated by an asterix). **h** A single AP was evoked by the ramp current injection (0 to 20 pA). **i** Trains of APs could be initiated with extend current injection (20 pA, 10–360 ms at 50 ms increment). **j** APs were evoked by step currents injection (−10 to 40 pA). **k** Voltage-gated K+ and Na+ currents were detected following the depolarizing voltage steps (−90 to +60 mV at 10 mV increment). **l** N-methyl-D-aspartic acid (NMDA)-gated currents could also be evoked in a neuron (hold at −70 mV) by exogenous NMDA (1 mM) application.

proliferation (Fig. 1k). Our results suggest that FGF signalling is mediated by both the MAPK (bar 7) and PI3K (bar 8) axis, as inhibition of either pathway significantly reduced cNESC numbers (Fig. 1k). Moreover, AKT (bar 9) inhibition did not interfere with the cell proliferation in 4F, suggesting that high activity of AKT is not required for PI3K signalling in cNESC maintenance (Fig. 1k).

In summary, we determined that FGF signalling through the MAPK and PI3K pathways and inhibition of GSK3 and SMAD signalling are required for cNESCs maintenance, while additional developmental signals like NOTCH, EGF and SHH are not required.

**Multipotent cNESCs differentiate to glutamatergic neurons.** Next, we tested whether the differentiation potential of the cNESCs was maintained after prolonged culture. We differentiated cNESCs maintained in vitro for 75 days. After 30 days of differentiation, cultures contained β3-tubulin (TUBB3)-positive neurons (approx. 60% of cells) that also expressed MAP2 and NEUN (approx. 40% of cells) (Fig. 2a, b), and Synaptophysin (SYP) positive presynaptic complexes were detectable on MAP2-positive dendrites (Fig. 2b) indicating that cNESC derived neurons started to mature. After neuronal differentiation cNESCs could also differentiate into GFAP-positive astrocytes, and O4-positive oligodendrocytes (Fig. 2a, c). Thus, confirming that cNESCs maintain multipotency during long-term culture.

To further characterise the types of cortical neurons produced, we analysed their neurotransmitter expression. The majority of cells were glutamatergic projection neurons (vesicular glutamate transporter type 1 (VGLUT1)-positive) (Fig. 2d). GAD67 positive and vGAT positive interneurons comprised around 10 and 2%, respectively, of the total number of neurons (Fig. 2e, f).

Next, we determined whether cNESC-derived neurons were electrophysiologically active mature cells. After 45 days of in vitro

Next, we tested if 4F was able to support the proliferation of a single cNESC in the absence of additional signals that could be produced by adjacent cells. We plated cNESCs previously maintained in 4F at clonal cell density in either 4F or FGF alone, without supporting cells or conditioned media. cNESCs in FGF alone rarely formed colonies (1 in 3000 cells, n = 3) both at low and high oxygen levels (3 and 20%; Fig. 1i and Supplementary Fig. 1f). In contrast, single cNESCs in 4F supported single cell derived colony formation at both low and high oxygen levels at an efficiency of around 1%, demonstrating that four pathways can maintain the neuroepithelial specification of a single cNESC (Fig. 1i and Supplementary Fig. 1f).

4F did not include developmental factors previously shown to support neural progenitor cell maintenance in vitro, such as EGF, NOTCH ligands or SHH[21,30]. Therefore, we tested their role in cNESC proliferation as we had observed a decrease in the proliferation of cNESCs in FGF or FGF with EGF (Fig. 1j, k; bars 14,15). Although cNESCs can activate all three receptors (Supplementary Fig. 1g–j), the inhibition of these developmental signalling pathways had no effect on the proliferation of cNESCs in the presence of the 4F (Fig. 1k).

Our signalling screen showed that a change in the activity of any factor in the 4F condition affects cell proliferation even when the other three components are present (Fig. 1k); activation of ACTIVIN A signalling, removal of GSK3 inhibitor or inhibition of the FGF receptor by PD173074 all caused a reduction in cell

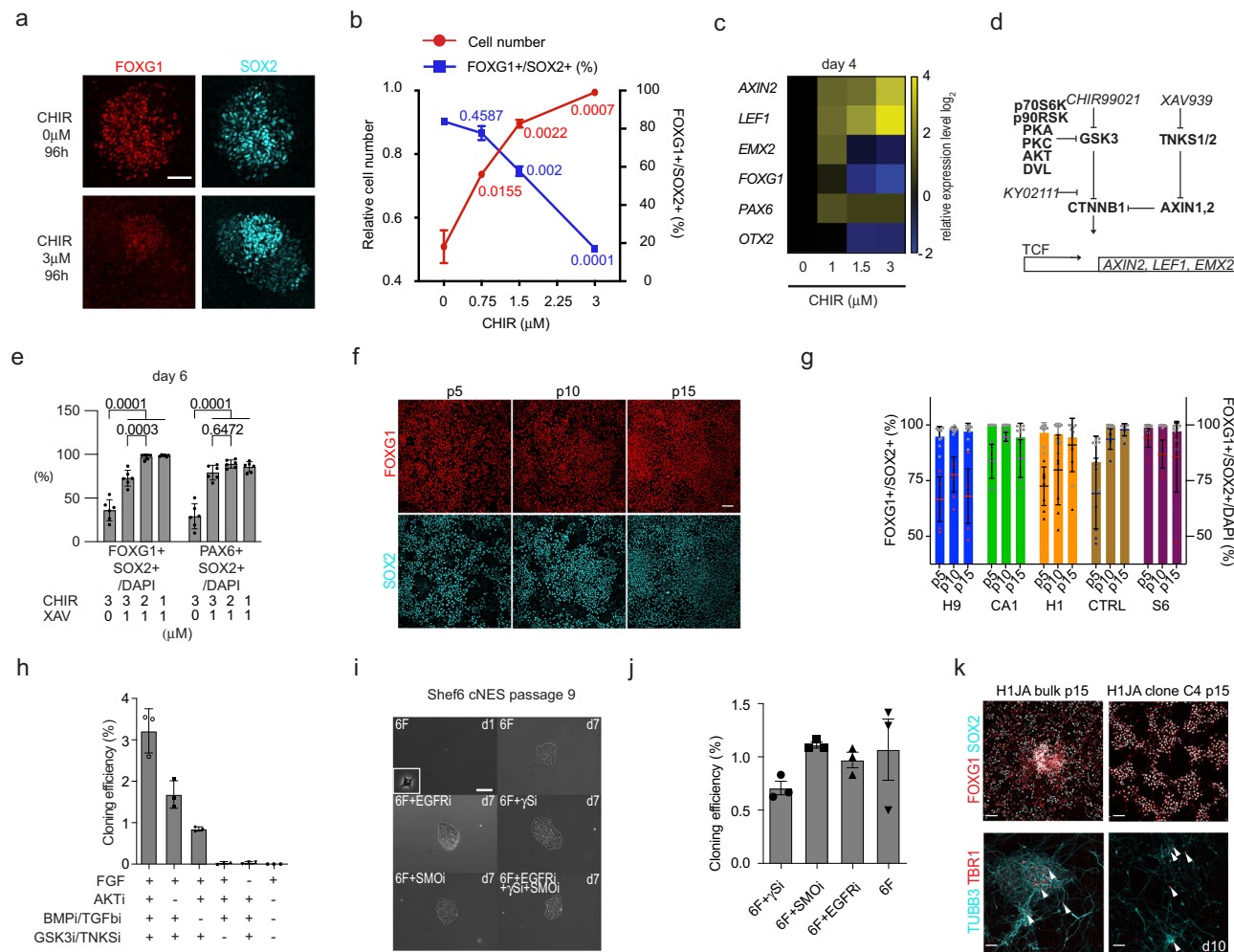

**Fig. 3 Low CTNNB1 activity maintains cortical specification of cNESCs. a** GSK3 inhibition reduces FOXG1 expression in SOX2 positive NES cells. The experiment was repeated with 3 biologically independent cell lines. Scalebar: 25 μm. **b** GSK3 inhibition increases cNESC numbers and decreases FOXG1 positive cell number ratio over a 96-h treatment in the presence of FGF with TGFbR and BMPR inhibition. ($n = 3$ independent experiments, data are presented as mean ± SD, 1-way ANOVA, Tukey's test, p values are indicated for each comparison to 0 μM CHIR). **c** Q-RT-PCR comparison of mRNA levels of selected dorsal forebrain specific genes and direct CTNNB1 target genes *AXIN2* and *LEF1* after 96 h treatment of H1 cNESCs with various concentrations of GSK3 inhibitor (CHIR) in 4F media, normalised to 0 μM treatment. ($n = 3$ biologically independent experiments, data are presented as mean). **d** Schematic of protein and chemical regulators of GSK3 and CTNNB1. Proteins are in bold, chemicals are in italic. **e** Quantification of FOXG1 and PAX6 positive SOX2 postive H1 cNESCs after 6 days culture in 4F media with various concentrations (μM) of GSK3 inhibitor (CHIR) and Tankyrase inhibitor XAV939 (XAV). ($n = 6$ independent samples, data are presented as mean ± SD, 2-way ANOVA, Tukey's test). **f** Immunofluorescent labelling of FOXG1 in H1JA cNESCs cultured in 6F condition from multiple passages. Scalebar: 25 μm. **g** Quantification of immunofluorescent labelling of FOXG1 in five independent hPSC derived cNESC lines from multiple passages. Bars and grey symbols show percentage of FOXG1 positive NES cells, lines and coloured symbols show percentage of FOXG1 and SOX2 positive cells of all cells. ($n = 5$ biologically independent cell lines, data are presented as mean ± SD). **h** Quantification of colony formation capacity of H1 cNESCs in 6F, 6F-AKTi, 6F-BMPi/TGFbi, 6F-GSK3i/TNKSi, 6F-FGF, FGF only. ($n = 3$ biologically independent experiments, data are presented as mean ± SD). **i** CNESCs (S6) form colonies in 6F condition in the presence of EGFR, SMO or γS inhibitor or the combination of them. The experiment was repeated with 3 biologically independent samples. Scalebar: 50 μm. **j** CNESCs (H1JA) were cultured in 6F media and plated at clonal density (200cells/cm2). Addition of NOTCH inhibitor DAPT, SMO inhibitor CYCK or EGFR inhibitor PD15 did not inhibit colony formation of cNESCs ($n = 3$ biologically independent cell lines, data are presented as mean ± SEM). **k** Top: Single cell derived clones of cNESCs H1JA were cultured in 6F media. Dorsal forebrain marker FOXG1 was co-labelled with SOX2 neural marker by immunofluorescence in bulk cultures (left) and a clone of H1JA cNESCs (right, H1JA Clone C4). Bottom: H1JA and Clone C4 cNESCs could differentiate to TBR1 positive neurons (TUBB3) in ten days. Arrowheads indicate TBR1 and TUBB3 positive neurons. Scalebar: 20 μm. The experiment was repeated with 5 biologically independent samples. Source data are provided as a Source Data file.

differentiation, we observed spontaneous action potentials in around 50% of the cells (Fig. 2g). These cells could be further stimulated to fire action potentials by applying ramp currents (Fig. 2h). When constant currents ($+20$ pA) were held for an extended period (10–360 ms) (Fig. 2i) or increasing the current steps from $-10$ pA to $+40$ pA up to 1000 ms (Fig. 2j), trains of action potentials were recorded. With increased depolarizing voltage steps, we were able to detect both fast voltage-gated Na$^+$ currents and subsequent voltage-activated K$^+$ currents (Fig. 2k), confirming the presence of necessary ion channel components for action potentials. Moreover, addition of glutamate receptor agonist *N*-methyl-D-aspartic acid (NMDA, 1 mM) and consequent activation of postsynaptic receptors resulted in the induction of excitatory membrane currents (Fig. 2l). Together,

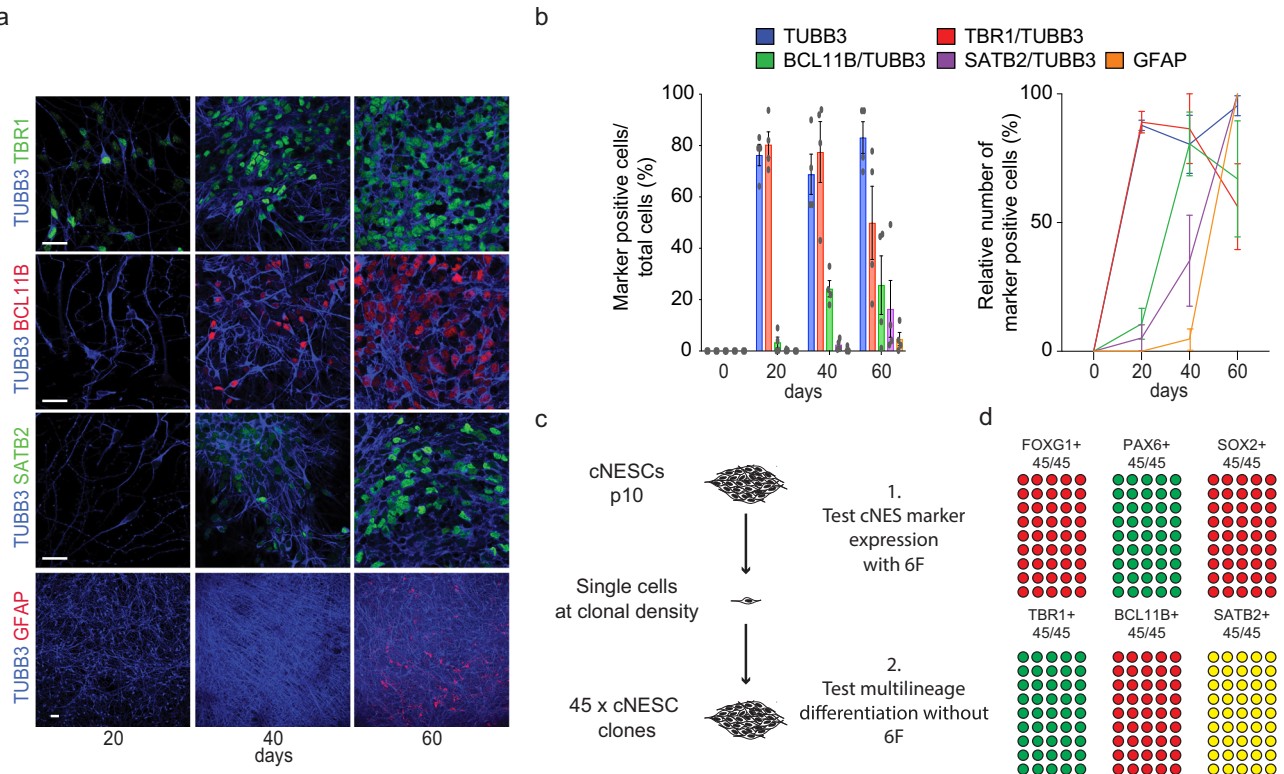

**Fig. 4 cNESCs in 6F maintain cortical developmental potential. a** Cortical layer specific TBR1-BCL11B-SATB2, neuron specific TUBB3 and astrocytic GFAP marker labelling of differentiated SHEF6 derived cNESCs (passage 12) at various time points. Scalebar: 10 µm. **b** Quantification of cortical layer specific marker positive neurons and astroglial cells in cNESC (passaged for at least 12 times in 6F) differentiating cultures (H1JA, CA1J, S6, H1JA-C4), left panel shows percentage of marker positive cells, right panel shows normalized numbers, 100% is the highest value during the time course (Mean values with SEM, $n = 4$). **c** Schematics of isolation of clonal cNESCs in 6F condition (left). **d** Quantification of immunofluorescent labelling of indicated cNESC markers (top) and differentiation markers of various cortical layers (bottom) in 45 clonal cNESC populations. Each circle represents a cNESC clone, filled circles represent marker positive clones. TBR1 and BCL11B expression was tested at day 30 and SATB2 was tested at day 60 of differentiation. Source data are provided as a Source Data file.

these results demonstrate the functional maturation of neurons derived from cNESCs.

**GSK3 and CTNNB1 regulation maintains proliferative cNESCs.** Our data showed GSK3 inhibition improved cNESC proliferation in FGF alone or in the 4F (Fig. 1g), however early anterior neuroepithelium specification requires low CTNNB1 activity[31]. Therefore, we tested whether forebrain specification is maintained in the presence of GSK3 inhibition. Gradually increasing GSK3 inhibition (from 0 to 3 µM GSK3i/CHIR) in cNESCs caused an increase in the proliferation rate of cNESCs and an immediate decrease in the number of FOXG1-positive cNESCs (Fig. 3a, b and Supplementary Fig. 2a). Increasing levels of GSK3 inhibition were associated with an increase in the expression of canonical WNT/CTNNB1 target genes *AXIN2* and *LEF1*, and a decrease in *EMX2*, *FOXG1* and *OTX2* mRNA levels, whereas *PAX6* mRNA level did not change (Fig. 3c). Long-term maintenance of cNESCs in the 4F media reduced the potential of the cells to differentiate to TBR1 and BCL11B (CTIP2) positive deep layer neuronal subtypes compared to direct differentiation of cNESCs without culture in 4F (Supplementary Fig. 2b). Together, these results suggested that low levels of both GSK3 and CTNNB1 activity are needed to maintain dorsal forebrain specification in cNESCs.

We aimed to establish the conditions that support cNESC proliferation while maintaining dorsal forebrain specification by regulating the WNT/CTNNB1 activity by lowering the GSK3 activity in combination with low CTNNB1 transcriptional

activity. We tested two different chemical inhibitors of CTNNB1 signalling, XAV939 and KYO2111 (Fig. 3d). XAV939 inhibits Tankyrase activity, which leads to an increase in AXIN2 levels and reduces CTNNB1 level in the nucleus[32]. KY02111 was shown to inhibit CTNNB1 function downstream of GSK3 during cardiomyocyte specification by an unknown mechanism[33]. High GSK3 inhibition (3 µM) rapidly increased mRNA expression of *LEF1* in cNESCs (Supplementary Fig. 2c). In comparison to GSK3 inhibition alone, GSK3 inhibition together with Tankyrase inhibition by XAV939 reduced the transcription of *LEF1* to control levels while the addition of KY02111 had no effect. Therefore, we tested whether dual inhibition of GSK3 and CTNNB1 transcriptional activity could maintain FOXG1 and PAX6 expression in SOX2 positive cNESCs after neural induction. Six days after replating the cells with or without high GSK3 inhibition (3 µM) reduced the number of both FOXG1 and PAX6 positive cNESCs, and this effect was compensated by the presence of 1µM XAV939 (Fig. 3e). Decreasing the concentration of GSK3 inhibitor to 2 or 1 µM in the presence of 1µM XAV939 preserved the expression of FOXG1, PAX6 and SOX2 in most of the cells. Based on the high proliferation rate and sustained marker expression of cNESCs at 2µM CHIR and 1µM XAV (C2X1), we decided to use this condition for further experiments.

It has previously been reported that neural progenitor proliferation and survival in frog and mouse embryos are regulated by sub-cellular localization of FOXG1 and FOXO proteins via AKT-mediated phosphorylation[34,35]. We found that FGF signalling activates both MAPK and PI3K/AKT pathways in

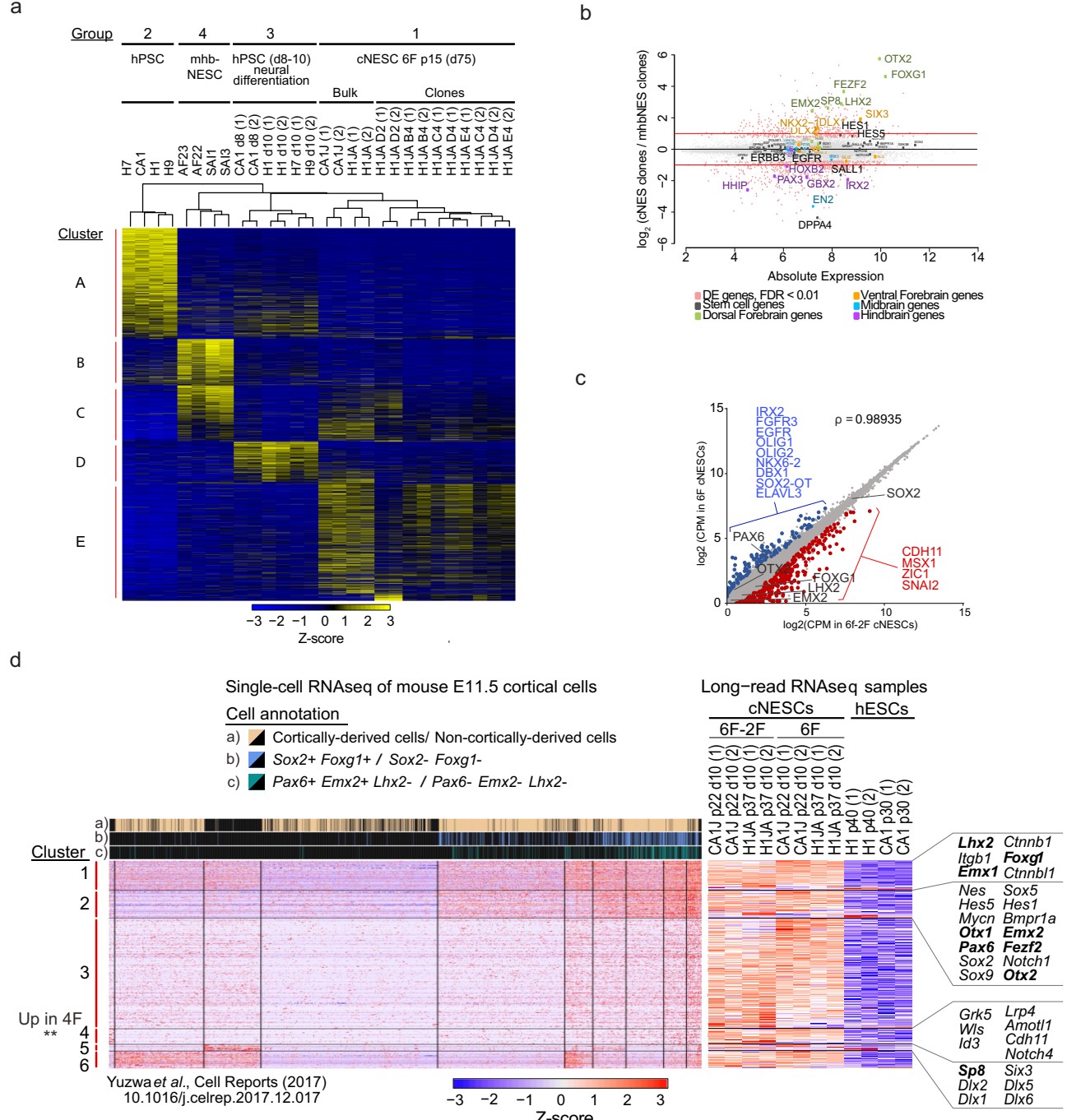

**Fig. 5 Sustained cortical gene expression in cNESCs. a** Heatmap of differentially expressed genes among four biological groups of samples. Gene expression levels are presented by Z-score. **b** MA plot representation of differentially expressed genes between cNESC and mhbNESC clones. Differentially expressed genes are highlighted in light red, selected stem cell (grey), dorsal forebrain (green), ventral forebrain (orange), midbrain (blue) and hindbrain (purple) related genes are named and highlighted with corresponding colour. **c** cNESCs (H1JA and CA1J) were cultured in 6F and compared to the same passage cells switched from 6F to 6F-2F condition for 10 days. Dot plot of gene expression in 6F-2F versus 6F cNESCs. Genes significantly enriched are highlighted (6F-2F: red, 6F: blue) and genes of interest are identified. Pearson correlation coefficient is indicated (ρ). **d** Gene expression clustering heatmaps for single-cell RNA sequencing (scRNAseq) from mouse cortex at embryonic day 11.5 (Yuzwa et al.[37]) (left) and lrRNAseq of cNESCs and hESCs (right), for cNESC marker genes obtained from literature and differential expression analysis. Single cells are annotated according to a) their origin and b-c) expression of in vivo cortical progenitor markers (top). Genes of interest are highlighted, and dorsal forebrain markers are presented in bold.

cNESCs. This was demonstrated by the phosphorylation of ERK and FOXO1/P70S6K and by the reduction in FOXO1 and P70S6K phosphorylation following inhibition of AKT kinase activity (Supplementary Fig. 2d). As we found that AKT inhibition does not interfere with cNESC maintenance (Fig. 1k),

we inhibited AKT to enhance FOXG1 function. cNESCs cultured in 4F with Tankyrase and AKT inhibitor (termed 6F for 6 factor condition) showed sustained expression of FOXG1 in cNESCs.

Next, we aimed to determine whether cNESCs cultured in 6F would maintain the expression of cortical markers FOXG1,

OTX1/2, PAX6 during extended culture. cNESCs cultured in 6F for 75 days (15 passages or 50 population doublings) maintained expression of FOXG1 in all five hPSC lines tested (Supplementary Table 2). No decrease in the ratio of FOXG1/SOX2 positive cells (Fig. 3f, g) was observed. Moreover, the expression of high levels of OTX1/2 (caudo-lateral cortex) or PAX6 (dorso-frontal cortex) was maintained in the SOX2 positive cNESCs from all hPSC lines (5/5; Supplementary Fig. 2e, f). The cNESCs proliferated consistently in the 6F condition and maintained a normal karyotype after more than 180 days (approximately 120 population doublings) in culture (Supplementary Fig. 2g). In addition, we found that most of the cells in the culture were actively proliferating. There was a low ratio of P21/Cyclin-dependent kinase inhibitor 1 positive quiescent cells among the total cells in the culture (Supplementary Fig. 2h, i).

Next, we tested whether 6F could support the formation of single cell derived colonies. Three percent of cNESCs cultured in 6F at a low density (200cells/cm2) formed colonies (Fig. 3h, i and Supplementary Fig. 2j, k). Removal of either AKT or BMP and TGFβ inhibitors resulted in colony formation, albeit at a reduced level (Fig. 3h). Similarly, to the 4F condition, colony formation was also seen with inhibition of NOTCH, EGFR or SMO receptors (Fig. 3i, j).

We then tested if this sustained cortical specification also preserved the potential of the cNESCs to differentiate into TBR1 positive neurons. We found that both bulk cNESCs and single cNESC-derived clones cultured in 6F maintained the expression of dorsal forebrain progenitor cell markers and retained the capacity to generate TBR1 positive neurons (Fig. 3k, l and Supplementary Fig. 2k, l). However, when single cell derived clones were cultured in 6F media lacking the Tankyrase inhibitor (XAV939), an increase in CTNNB1 activity led to increased expression of AXIN2, LEF1, PAX6 and NGN1 genes and reduced expression of FOXG1, OTX2, DLL3, DLL1 and ASCL1 mRNA (Supplementary Fig. 2m) compared to cNESCs in complete 6F. These results suggest that cNESCs require a balanced regulation of GSK3, CTNNB1 and AKT activity to maintain a dorsal telencephalic fate and that the regulation of these pathways supports the formation of single cell derived colonies with early developmental potential.

**cNESCs preserve cortical differentiation programme**. During development, the differentiation of glutamatergic projection neurons follows an "inside first, outside last" pattern, which results in the formation of the six layers of the neocortex. To determine whether 6F cNESCs could recapitulate this pattern, we examined the potential and timing of cNESC differentiation to lower- and upper- layer neurons. Strikingly, we saw the layer 6, TBR1 positive neurons appear first (Fig. 4a, b), followed by layer 5 specific BCL11B positive neurons (Fig. 4a, b) and finally the appearance of upper layer specific SATB2 positive neurons (Fig. 4a, b). The generation of neurons was followed by the emergence of GFAP positive astrocytes (Fig. 4b). This showed that 6F cNESCs preserve the "inside first, outside last" embryonic pattern of differentiation (Fig. 4b, right panel).

To determine whether individual cNESCs can retain multilayer cortical differentiation potential in 6F, we performed long-term culture of subcultured cNESC colonies. We plated passage 10 cNESCs (Shef6) at clonal density in a 10 cm dish in 6F medium and subcultured 48 colonies after 10 days in 96 well plate (Supplementary Fig. 3a). 45 out of 48 colonies could be plated and expanded in the 96 well plate. To determine whether the Shef6 cNESC clones maintained dorsal forebrain specification in 6F we examined the FOXG1, PAX6 and SOX2 expression. All cNESC clones (45/45) expressed SOX2, FOXG1 and PAX6

markers (Supplementary Fig. 3b), demonstrating the maintenance of dorsal forebrain specification in the 6F condition. To assess whether 6F cNESC clones maintained multilayer specific differentiation potential, we differentiated cells by withdrawing 6F and looked for the presence of TBR1 positive or BCL11B positive deep layer cells, and SATB2 positive upper layer cells. We found that all clones (45/45) had the potential to differentiate into TBR1 positive layer 6 and BCL11B positive layer 5 cells in 30 days (Supplementary Fig. 3c). After sixty days of differentiation, we were also able to identify SATB2 positive upper layer cells in all differentiating clones (Supplementary Fig. 3c), demonstrating the maintenance of multilineage potential of cNESCs in the 6F condition.

**cNESCs maintain embryonic cortical gene expression**. The 6F condition preserved key marker expression and differentiation potential of early cortical neuroepithelial progenitors. To confirm the forebrain-specific gene signature of cNESCs, we compared the transcriptome of 6F cNESCs (Group 1; H1JA-derived clones and CA1J and H1AJ-derived bulk cultures, Fig. 3k, Supplementary Fig. 2k, l); hPSC differentiated to dorsal forebrain neuroepithelium for 8–10d without culture in 6F condition (Group 2; H1, H7, H9 and CA1); mid-hindbrain specified NESCs (mhbNESCs) (Group 3; derived in FGF and EGF from hPSCs -AF22, 23[18] or from human embryos - SAI1,3[36]); and hPSCs (Group 4; H1, H7, H9, and CA1), (Fig. 5a). 6F cNESCs showed different gene expression signatures from mhbNESCs and hPSCs (Fig. 5a) and were most similar to dorsal forebrain specified cultures (Group 2). mhbNESCs (Group 3) clustered differently from both hPSCs (Group 4) and all dorsal forebrain specified samples (Group 1) (Fig. 5a).

We then analyzed gene clusters that were specifically expressed in cNESCs. Our analysis revealed that 6F cNESCs (Group 1) expressed genes related to neural differentiation, neurogenesis, axon development, cell-cell signalling and cell adhesion. We detected increased expression of genes related to forebrain development, cilium morphogenesis and negative regulation of both neurogenesis and gliogenesis (Cluster E, Supplementary Fig. 4a). Since we were able to detect the enrichment of forebrain development-specific genes in our transcriptome analysis of 6F cNESC samples, we specifically examined genes related to forebrain and mid-hindbrain specification and development in 6F cNESCs and mhbNESCs (Group 3, Supplementary Fig. 4b-d). Both cNESCs and mhbNESCs had similar expression levels of neural progenitor genes SOX1, SOX2, NES, NOTCH1, PAX6 (Fig. 5b). Strikingly, 6F cNESC clones showed upregulation of forebrain-specific genes, including OTX2, FOXG1, FEZF2, SP8, EMX2, SIX3 while mhbNESCs showed higher expression of DPPA4, PAX3, EN2, GBX2, IRX2, HOXB2, GLI2 and HHIP, thus demonstrating distinct gene expression patterns that reflect their respective anatomical regions (Fig. 5b, Supplementary Fig. 4b–d).

To further characterize the identity of 6F cNESCs, we performed long-read RNA sequencing of passage 20+, 6F cNESCs (H1JA and CA1J). To specifically understand changes in gene expression that were associated with the AKT and Tankyrase inhibition, we cultured cNESCs in 6F and then removed these two factors (6F-2F) for 10 days prior to sequencing. We found sustained expression of cortical neuroepithelial markers SOX2, FOXG1, OTX2, EMX2, LHX2 in both the 6F and 6F-2F conditions (Fig. 5c). However, cNESCs in the 6F-2F condition had reduced expression of 254 genes such OLIG1, OLIG2, NKX6-2, FGFR3 expressed in the ventral forebrain and later in oligodendrocyte lineage and elevated expression of 70 genes such as neural crest lineage genes like CDH11, MSX1, SNAI2 indicating a small, but rapid change of the transcriptome after removal of the AKT and Tankyrase inhibitors.

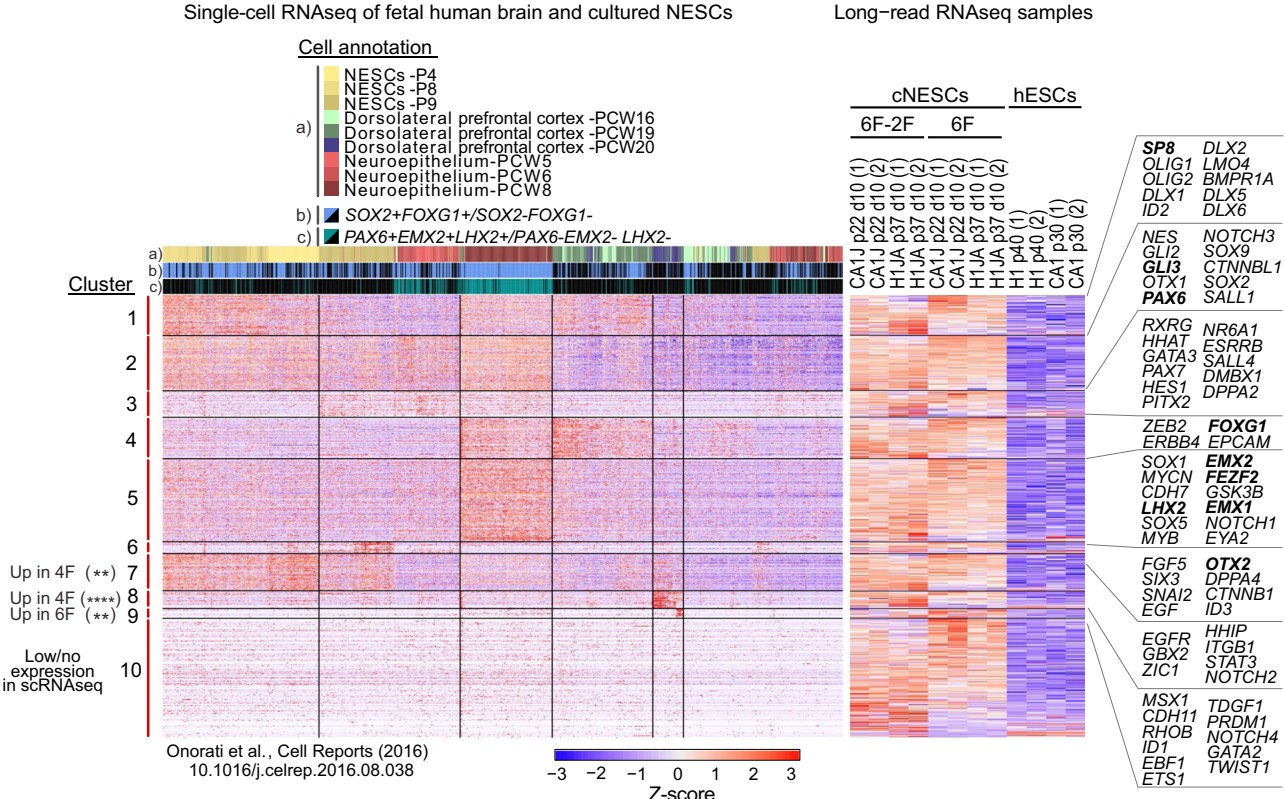

**Fig. 6 Human developmental gene expression in cNESCs.** Gene expression clustering heatmaps (as Z-score of log2(CPM + 1)) for single-cell RNA sequencing (scRNAseq) from foetal human brain tissue and cultured NESCs (Onorati *et al.*, 2016[38],) (left) and lrRNAseq of cNESCs and hESCs (right), for cNESC marker genes obtained from literature and differential expression analysis. Single cells are annotated according to a) their origin and b-c) expression of in vivo cortical progenitor markers (top). Genes of interest are highlighted, and dorsal forebrain markers are presented in bold. P = passage, PCW = post-conception week. Source data are provided as a Source Data file.

To determine how the gene expression of the cNESCs in the 6F and 6F-2F conditions compared to bona fide embryonic cortical neuroepithelial progenitors, we used the single cell transcriptomic data from day 11.5 mouse embryonic cortices or gestational week 5 to 20 human cortices[37,38]. Both datasets contained progenitor cells that expressed *SOX2, FOXG1, EMX2, PAX6* and *LHX2* genes. This gene expression signature was used to identify dorsal forebrain/cortically specified cortical neuroepithelial progenitors (Supplementary Fig. 5a, b).

To determine the gene signature of cNESCs we examined the differentially expressed genes in 6F and 6F-2F cNESCs when compared to isogenic hESCs. To confirm dorsal forebrain cortical neuroepithelial identity, we compared cNESCs gene expression to cortex-derived cells (i) from the Yuzwa et al. mouse embryonic brain dataset, and then specifically to *Foxg1/Sox2* expressing or (ii) *Pax6/Lmx2/Emx2* (iii) expressing subpopulations. The genes enriched in *Foxg1/Sox2* (ii) and *Pax6/Lmx2/Emx2* (iii) were also highly expressed in 6F and 6F-2F cNESCs (Fig. 5d, Supplementary Fig. 6a). In addition, cNESCs showed high expression of dorsal forebrain genes *EMX1, FEZF2, SOX2, HES1, HES5, MYCN*. These were associated with transcriptional regulation, neural stem cell maintenance, negative regulation of neuronal differentiation and forebrain radial glia cell differentiation (Clusters 1 and 2, Supplementary Fig. 7a). When we compared these clusters of genes in the mouse cNESCs we found no significant difference in the global expression between the 6F and 6F-2F conditions, which confirmed our previous findings in panel A (Fig. 5c, Supplementary Fig. 8a). We found an upregulation of genes in Cluster 4 in the 6F-2F condition, including genes expressed in the neural crest lineage such as *Id3, Notch4, Cdh11*, but altogether these genes

were not associated with any gene ontology (GO) term or KEGG biological pathway (showed a Supplementary Fig. 7a).

We then compared 6F and 6F-2F cNESCs to human embryonic and foetal cortices. Similar to the analysis performed for mouse cortical neuroepithelial progenitors, we examined the differentially expressed genes in 6F and 6F-2F cNESCs when compared to isogenic hESCs. To confirm dorsal forebrain cortical neuroepithelial identity, we compared cNESCs gene expression to cultured NESCs, gestational week 5, 6, 8, 16, 19, 20 human cortical progenitors (i), and then specifically to *FOXG1/SOX2* expressing (ii) or *PAX6/LMX2/EMX2* (iii) expressing subpopulations (Fig. 6). The genes enriched in *FOXG1/SOX2* (ii) and *PAX6/LMX2/EMX2* (iii) were also highly expressed in 6F and 6F-2F cNESCs (Fig. 5a, Supplementary Fig. 6b). These highly expressed genes included, *FOXG1, SOX1, EMX2, FEZF2, EMX1, and LHX2*, and were not expressed at earlier (gestational week 5 or 6 or later gestational stages (gestational week 16, 19 or 20). When we compared the expression of these genes in the 6F and 6F-2F cNESCs, we could find no significant difference (Clusters 4 and 5, Fig. 6, Supplementary Fig. 8b). cNESCs in the 4F condition significantly elevated the expression of genes such as *ZIC1, HHIP, ITGB1, NOTCH2, NOTCH4, MSX1 CDH11, GATA2* expressed in the neural crest lineage and associated to biological processes related to neural crest cellular function (Cluster 7 and 8, Fig. 6, Supplementary Figs. 7b, 8b). We found genes upregulated in the 6F condition, but we could not find enriched association to any known biological function (Cluster 9, Fig. 6, Supplementary Figs. 6b, 7b, 8b).

Altogether, our transcriptome analysis verified that 6F cNESCs show a specific dorsal forebrain identity that is different from

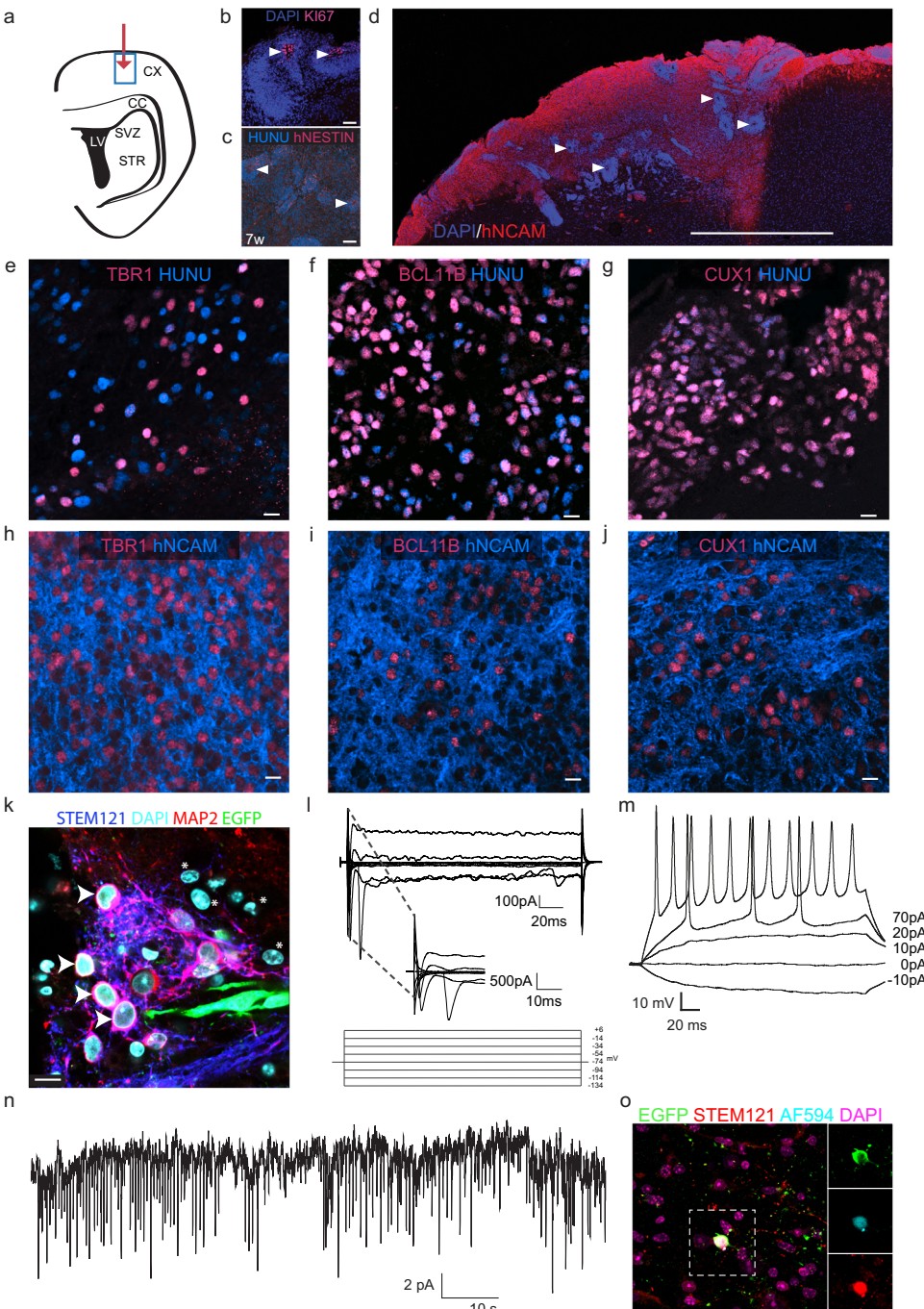

mid-hindbrain specified NESCs in vitro, and a gene expression signature that is consistent with embryonic cortical progenitors in the mouse and human tissue. Moreover, we find the removal of AKT and Tankyrase inhibitors is associated with changes in gene expression related to ventral forebrain and neural crest cell lineages.

**Transplanted cNESCs execute a cortical programme in vivo**. To determine whether cNESCs would follow their early developmental differentiation programme in vivo, we transplanted undifferentiated, proliferating 6F cNESCs into the cerebral cortex of 9-week-old mice (Fig. 7a). Seven weeks post-transplantation, we saw formation of rosette-like polarised structures (indicated by arrowheads) that contained KI67 positive cycling progenitors (Fig. 7b) and NESTIN positive cells (Fig. 7c). By thirteen weeks

cNESCs differentiated into a large number of hNCAM positive neurons (Fig. 7d). When we analysed these cells for cortical layer specific markers, we found TBR1 and BCL11B positive deep layer cells and CUX1 positive upper layer cells in distinct, non-overlapping populations of cells (Fig. 7e–g, Supplementary Fig. 9a–c). Both deep and upper layer specific human cells gave rise to layer specific, hNCAM positive neurons (Fig. 7h, i, Supplementary Fig. 9d–f). As many cortical neurons project to the contralateral hemisphere via the corpus callosum, we looked for the expression of hNCAM in the hemisphere contralateral to the transplant. We noticed the presence of several hNCAM processes and the absence of Human Nuclear Antigen (HuNu) positive nuclei in the contralateral corpus callosum, suggesting that cNESC-derived cortical neurons can project long distances (Supplementary Fig. 9g). These results show that, not only

**Fig. 7 Transplanted cNESCs can complete embryonic cortical differentiation potential in the postnatal brain. a** Schematic of 6-week-old mouse coronal brain section indicating the injection site (red arrow) and analysed area (blue rectangle). **b** After 7 weeks post transplantation polarized rosette structures (arrowheads) of transplanted cNESCs (H1JA, CA1J) contain proliferating KI67 positive human cells. The experiment was repeated with 2 biologically independent cell lines. Scalebar: 50 µm. **c** NESTIN positive human progenitors were found in the grafts at 7 weeks. Arrowheads indicate rosette structures. Scalebar: 50 µm. **d** Human NCAM expressing neurons are abundant in grafts of cNESCs seven weeks after transplantation. Scalebar: 1 mm. **e**–**j** Transplanted human cNESCs differentiated in situ to TBR1 **e**, **h**, BCL11B **f**, **i** and CUX1 **g**, **j** positive HuNu and hNCAM expressing cells by 13 weeks after transplantation. The experiment was repeated with 2 biologically independent cell lines (H1JA, CA1J). Scalebars: 10 µm. **k** Immunofluorescent detection of differentiated human cells in the NSG mouse cortex 12 weeks after transplantation. Cytoplasmic marker STEM121 (blue) identifies human neurons (H1JA-APCE) that express MAP2 (red) and EGFP (green) indicated with arrowheads and surrounded by mouse cells that are negative for all three markers indicated by asterisks. Several human MAP2 STEM121 double positive neurons were EGFP dim or negative and can be identified only with immunolabelling (see Supplementary Fig. 9H). Nuclei were labelled with DAPI. The experiment was repeated with 3 animals. Scalebar: 10 µm. **l** Voltage gated sodium and potassium currents can be recorded from EGFP positive neurons (H1JA-APCE) in the cortical slices of NSG mice 12 weeks after transplantation. Voltage values are indicated under the traces. **m** Human EGFP positive neurons (H1JA-APCE) surrounded by mainly host rodent cells can fire action potentials after current injection in 80% of patched neurons (4/5) 12 weeks after transplantation. Current steps are indicated next to the traces. **n** Spontaneous EPSCs could be detected in 40% (2/5 neurons) of EGFP positive human neurons (H1JA-APCE) surrounded by mainly host rodent cells 12 weeks after transplantation. **o** Post hoc immunofluorescent labelling of recorded cells 12 weeks after transplantation. EGFP positive cells (H1JA-APCE) were filled with ALEXA 594 dye from the recording internal solution. ALEXA 594 (blue) and EGFP (green) positive cells were also labelled for STEM121 (red) confirming the human origin. Surrounding cells are triple negative host cells. Nuclei were labelled with DAPI (purple). The experiment was repeated with 3 animals. Scalebar: 10 µm.

cNESCs maintain their potential to generate both deep and upper layer neurons in vitro but also in vivo.

We then determined whether these neurons were able mature in the host tissue and synapse with other neurons. We labelled cNESCs with EGFP knocked-in to the human AAVS1 locus and transplanted the cells to adolescent mice. 12 weeks after injection we found EGFP positive cells in the grey matter of the mouse cortex that expressed the human specific marker STEM121 and MAP2 (Fig. 7k, Supplementary Fig. 9h). Patch-clamp recording of the transplanted EGFP positive neurons showed that these human neurons express voltage gated sodium and potassium channels and can generate action potentials (Fig. 7l, m). To determine whether the human EGFP positive neurons could connect to surrounding neurons, we recorded synaptic EPSCs from EGFP positive cells, and confirmed their identity by post hoc labelling for EGFP, STEM121 and AF594 from the patching internal solution (Fig. 7n, o). Altogether, these results suggest that transplanted human cNESCs can differentiate in the adolescent cortical tissue to mature neocortical neurons that can establish synaptic connections.

## Discussion

Cortical or dorsal forebrain neuroepithelial cells are the *founding* cells of the cerebral cortex. Before the onset of neurogenesis these cells divide symmetrically to expand the progenitor pool, but the regulation of their self-renewal is not clear. We investigated the necessary developmental signalling pathways for the maintenance of multipotent cortically-specified neuroepithelial stem cells (cNESCs) in vitro. By testing combinations of activators and inhibitors, we found that a combination of six factors (activation of FGFR together with inhibition of BMPR, TGFβR, GSK3, AKT and TNKS; termed 6F) can support the permanent self-renewal of human cNESCs. We showed that: (1) cNESCs can proliferate without changing their cortical marker expression; (2) cNESCs preserve the potential to differentiate to cortical neural cells following the embryonic developmental order; (3) single cNESCs can form colonies and preserve the developmental potential of the bulk population; (4) cNESCs can implement their developmental potential in the cortical tissue.

We found that FGFR activity is needed for human cNESC proliferation as it was shown in mouse forebrain neuroepithelium[6], and this is mediated by both MAPK and PI3K pathway, however AKT inhibition allows colony formation and proliferation. AKT has been shown to regulate the nuclear export

of FOXG1 in cortical progenitors[35], so reduced level of AKT and additional proteins will help the retention of FOXG1 in the nucleus and maintain cortical specification.

Our screen showed that a key component of the self-renewal of cNESCs is the regulation of GSK3 and CTNNB1 activity. Inhibition of GSK3 promotes proliferation and colony formation of human cNESCs, but the level of transcriptional CTNNB1 activity regulates the expression of essential dorsal forebrain specification genes FOXG1, EMX2, PAX6, and OTX1/2. Combined reduction of GSK3 and CTNNB1 transcriptional activity supports cNESC proliferation, colony formation and the maintenance of dorsal forebrain specification and differentiation potential. Removal of the CTNNB1 regulation (XAV939) causes increased nuclear activity that changes the transcriptome of the cNESCs. The continuous elevated CTNNB1 levels reduce FOXG1 expression and change the differentiation potential in the cNESCs as they no longer can form deep layer neurons positive for TBR1 and BCL11B. Studies on GSK3 function in the developing neural tissue showed the requirement of reduced kinase activity to inhibit neural crest formation and neuronal differentiation[39,40]. Several signalling pathways can alter the activity of GSK3, which acts as a hub protein[41,42] regulating the activity of several target proteins involved in cell cycle regulation, mRNA transcription and protein translation. GSK3 can interact with ERK to down-regulate SMAD mediated ACTIVIN A and BMP signalling[26,27]. We found that reduction of GSK3 activity in cNESC requires the inhibition of BMP and ACTIVIN A for the self-renewal of cNESCs. BMP inhibition in cNESCs maintains the expression of SOX1, an early marker of neuroepithelium[43], while inhibition of ACTIVIN A signalling supports cell proliferation and allows colony-formation of cNESCs. Conversely it has been shown that β-catenin (CTNNB1) transcriptional activity is low during the specification and maintenance of forebrain identity of the neuroepithelium[31] and this can be mediated by several inhibitory proteins (TLE2, TCF7l1, TCF3, HESX1, CTNNBIP1)[44–47]. In vitro chemical inhibition of Tankyrase by XAV939 can mimic the function of modulatory proteins in stem cells and balance CTNNB1 transcriptional activity[48].

NOTCH signalling has been shown to be important in brain development, but dispensable during the neuroepithelial expansion phase specifically in the forebrain[49]. The anterior neuroepithelium develops without *RBPJ* expression[50] and cleaved NOTCH intracellular domain is low in the apical progenitor population and accumulates in the nucleus of progenitors of the

forebrain from the start of neurogenesis at embryonic day 10.5[51]. The switch to NOTCH dependent state of progenitors at the start of neurogenesis is reflected rapid differentiation to neurons of *RBPJ* null embryonic forebrain cells. Our results define the necessary signals for the early cortical neuroepithelial progenitor self-renewal independently of NOTCH signalling and complements previous observations about this transient state in rodent cell cultures[23,52]. Human early embryonic neuroepithelial cells show heterogeneity at the transcriptome level and display differences to rodent cells[53]. The FOXG1, PAX6, SOX2 and TJP1 positive neuroepithelial cells are present from Carnegie stage 16 telencephalon before the differentiation of DCX positive deep layer cortical neurons. Transcriptomic analysis indicated dynamic changes of FGF, WNT, NOTCH and mTOR signalling in the neuroepithelial cells during gestational weeks 6 to 9[53]. Our study verified the requirement of FGF with low GSK3, CTNNB1 and AKT signalling is sufficient for FOXG1, PAX6, SOX2 positive cNESCs in vitro maintenance, even when NOTCH signalling is inhibited.

Hedgehog signalling specifies and supports the maintenance of ventral forebrain progenitors, but it is not required for the dorsal forebrain progenitors similar to our observations with the inhibition of SMO activity in cNESCs. The convergence of various signalling pathways to support progenitor cell maintenance in the cortex was shown by the capacity of Hedgehog signals to compensate the accelerated neurogenesis and progenitor pool depletion in the *RBPJ* null embryos[54].

We also found that cNESCs do not require EGFR signalling in the 6F condition. EGF induces elongated bipolar morphology in mouse cortical radial glia cells, allowing the cells to act as scaffold for migratory cells[55]. Long-term exposure of human forebrain neuroepithelial cells to FGF and EGF causes their transition to radial glia-like cells and differentiation into astrocytes[56]. EGFR-mediated signalling has been shown to be important for the gliogenic differentiation of neural progenitors[57], however multipotent neural progenitors in the cortex can function normally without EGFR activity during development[58]. Our data and previous observations suggest that early cNESCs does not require EGFR activity but when stimulated it causes radial glia-like cell proliferation and the loss of neurogenic cortical differentiation potential[7,18,19].

One of the key properties of self-renewing stem cells is the sustained differentiation potential over rounds of cell divisions. 6F cNESCs retain early cortical potency long-term and preserve the developmental order of corticogenesis. Once the cNESCs are allowed to differentiate by the removal of the 6F neurons of the TBR1 and BCL11B positive deep layers are produced first, followed by SATB2 and CUX1 positive upper layer neurons and later astrocytes.

The 6F condition supports the efficient formation of single cell derived colonies of SOX2 positive cNESCs that express FOXG1 and PAX6 dorsal forebrain progenitor markers indicating sufficient regulation of key signalling components to inhibit differentiation or regional repatterning. The individual cNESC clones could differentiate into deep (TBR1, CTIP2) and upper (SATB2) cortical layer specific cells demonstrating the ability of the early dorsal forebrain neuroepithelial cells to proliferate extensively without changing their differentiation potential. The identified 6F control of cNESC self-renewal supports the observation of Shen et al. that blood vessel endothelial cells can secrete factors that support the self-renewal of cortically specified neuroepithelial progenitors however, results on signals that can reduce GSK3, TGFBR, BMPR activity was not reported[8].

The lack of regenerative potential in the adult and ageing cerebral cortex is hypothesised to be due to the absence of stem cells with embryonic cortical potential and the inability of the tissue to regulate the generation or integration of new projection neurons. The transplantation of undifferentiated cNESCs into the young mouse brain demonstrated that cells with early embryonic developmental potential activate their differentiation programme in the cortical tissue environment and generate deep and upper layer projection neurons. The newly formed neurons could mature to electrophysiologically active cells in the host tissue and form functional synapses that receive inputs from other neurons similar to transplanted mouse embryonic neurons in the adult neocortex[59]. We observed that neuronal differentiation of cNESCs labelled with EGFP delivered by various methods correlated with transgene repression and silencing over time, therefore we cannot rule out that we could not identify a good fraction of mature and synaptically connected human neurons in the live brain slices. The in situ differentiation of transplanted human cNESCs and the survival of cortical neurons for at least 90 days in the adult brain provides an opportunity to investigate the regulatory mechanisms of cell lineage specification of early developmental stage cells and to investigate the regeneration or functional recovery of the adult cortex.

In summary, we present evidence that the six signalling pathway components control the self-renewal of early cortical neuroepithelial stem cells in vitro. The cNESCs preserve their early developmental potential and neuroepithelial cell state until the 6 signalling modulators are under control in vitro. Then, the transplanted human cNESCs differentiate in situ to projection neurons of the cerebral cortex in the young adult brain.

## Methods

All human stem cell work complied with the human stem cell guidance and was approved by the Stem Cell Oversight Committee of Canada and the Human Tissue Authority of United Kingdom, all animal work complied with regulations in Canada and United Kingdom and was approved by Mount Sinai Hospital Research Ethical board (Canada) and the Home office (UK).

**Human pluripotent stem cell culture and neural induction.** Human pluripotent stem cells (hPS) H1 (WAe001-A, WiCell), H9 (WAe009-A,WiCell), CA1 (Mount Sinai Hospital, Canada), SHEF6 (R-05-031,UK Stem Cell Bank), CTRL-2429 (University of Cambridge), 1.53E (Mount Sinai Hospital, Canada) were routinely cultured on mitomycin C treated mouse embryonic fibroblast feeder cells in KSR media with 10 ng/ml FGF2. KSR media contains 80% DMEM-F12 (GIBCO), 20% Knockout serum replacement (GIBCO), 0.1 mM 2-mercaptoethanol, 2 mM Glutamax, 0.1 mM Non-essential amino acids. Prior to neural differentiation hPSCs were transferred to geltrex (GIBCO) coated surface in the absence of feeders and were cultured in mTESR2 (Stem cell technologies) media. On the day of neural induction, hPSCs were replated on geltrex coated surface in N2B27 media supplemented with 10 μM SB431542 and 100 nM LDN193189 and 10 μM Y27632. Details of cell culture media components are listed in Supplementary Table 1.

**Neuroepithelial stem cell maintenance and differentiation.** Neural cultures were detached from Geltrex first on day 8–10 with Accutase or PBS-EDTA, and replated in neural maintenance media (NES) with 10 μM Y27632 at $3 \times 10^5$ cells per $cm^2$ on laminin (Sigma) coated surface. NES media contains 50% DMEM-F12, 50% Neurobasal, 0.1 mM 2-mercaptoethanol, 2 mM Glutamax, 1x N2 supplement, 0.05x B27 minus vitamin A supplement (GIBCO). 4F media is made by supplementing NES media with 10 ng/ml FGF2, 3 μM CHIR99021, 1 μM SB431542 or 50 ng/ml FST (Peprotech), 100 nM LDN193189 or 50 ng/ml NOGGIN. 6F media contains components of the 4F media with 100 nM K02288 (Selleckchem), 100 nM AKTiVIII (Selleckchem) and 75 nM MK2206 (Selleckchem), and 1–2 μM XAV939 (Selleckchem). XAV939 concentrations need to be adjusted to each cell line, 1 μM for H1, CA1, H9, CTRL hPSCd-erived cNESCs, 2 μM for Shef6 derived cNESCs. For oligodendrocyte differentiation cNESCs were cultured in 4F and cells were collected in clumps using EDTA-PBS. Floating cell clumps were cultured in 4F media for 2–3 days and the aggregates were transferred to Neural Differentiation I Media. Aggregates were cultured for 30 days in Neural Differentiation I Media with media changes every 4 days. Floating aggregates were plated onto Poly-D-lysin and laminin coated glass coverslips and cultured for 14 more days in Neural Differentiation I Media. Cultures were fixed with 4% Paraformaldehyde and stained with anti-O4 mouse IgM antibody. Details of cell culture media components are listed in Supplementary Table 1.

**Proliferation assay.** Proliferation of hNESCs cells was assayed by plating $1 \times 10^4$ cells per each laminin coated well of a 96 well plate in NES media. Cells were

allowed to attach and spread for 2 h and one volume of seeding media was complemented with equal volume of test media with 2x concentration test molecules. Tested molecules were used in dilution series and had the following highest final concentrations: ACTIVIN A (100 ng/ml, Peprotech), BMP4 (20 ng/ml, Peprotech) FST (50 ng/ml, Peprotech), NOGGIN (50 ng/ml, Peprotech), CHIR99021 (3 μM, Selleckchem), Cyclopamine-KAAD (200 nM, Millipore), DAPT (5 μM), EGF (100 ng/ml, Peprotech), FGF2 (10 ng/ml, Peprotech), LDN193189 (100 nM, Selleckchem), PD0325901 (1 μM, Selleckchem), PD153035 (2.5 μM), PD173074 (0.3 μM), PI-103 (0.5 μM), Wortmannin (0.1 μM), Purmorphamine (1 μM), SAG (100 nM), SB202190 (20 μM), SB203580 (20 μM), SB505124 (1 μM), SB431542 (10 μM), LY364947 (5 μM), TGFβ1 (100 ng/ml), Y27632 (10 μM), JNK-IN-8 (3 μM), SP600125 (1 μM). Cells were incubated with the indicated factor for 96 h. Total cell numbers were determined by Alamar blue cell viability assay (Life Sciences) based on cell number titration. Each value is the average of 6 technical replicates, average values were normalised to control condition (4F) and represented as relative values. All experiments were repeated using three cell lines (H1A, H9A, H7E). Details of chemical components, growth factors and cell culture media components are listed in Supplementary Tables 1, 3.

**Clonogenicity assay and single cell clone isolation.** H1 (passage 12) and S6 (passage 9) cNESCs were cultured in 4F or 6F medium as described in section 'Neuroepithelial stem cell maintenance and differentiation'. Three to four days after plating the cells were rinsed with PBS once and detached with 5-min Accutase digestion at 37 °C. The cells were collected with 10fold volume DMEM-F12 with 0.1% BSA and centrifuged at $300 \times g$ for 3.5 min. The pelleted cells were resuspended in 1 ml fresh 6F media and triturated by pipetting 1000 μl tip for 20 times with moderate force without generating bubbles. Cells were centrifuged with additional 5fold volume DMEM-F12 with 0.1% BSA at $300 \times g$ for 3.5 min. For clonogenicity assay the pelleted cells were resuspended in 1 ml fresh NES media with FGF2 at 20 ng/ml concentration and counted in a haemocytometer. Two hundred cells per cm² culture surface area were plated in poly-D-lysine and laminin coated plastic culture dishes. Two hours after the plating cells attached and spreaded well, and media was changed to test conditions. Fresh FGF was added daily at 20 ng/ml concentration without media change for the first 4 days until clones reached a colony size of 4 or more cells. Media was completely changed after 6 days. Cultures were fixed at indicated timepoints with 4% PFA solution for 10 min at room temperature. After rinsing with PBS the cell colonies were labelled with Cresyl violet solution for 5 min at room temperature and rinsed with water. The plates were scanned and colonies were counted with Volocity software. Occasionally colonies split to form 2–3 small colonies close to each other and separate from neighbouring colonies. The split colony clusters were counted as one.

During the in vitro differentiation assay at day 6 the wells were treated with accutase for 2 min, accutase was removed and 3 min later the cells were gently triturated with N2B27 to generate small clumps of cells. Cells were replated at 1:4 ratio in fresh 96 wells coated with poly-D-lysine and laminin. Cultures were fed every 4 days with 50% of the media volume replaced with fresh N2B27 media that contained 10 μM Forskolin and 20 ng/ml BDNF. Cultures were fixed at day 30 and 60 with 4% PFA solution for 10 min at room temperature.

For single cell cloning S6 cNESCs were plated in 6F media at 100 cells per cm² culture surface area in a 10 cm dish coated with poly-D-lysine and laminin. FGF was added daily at 20 ng/ml concentration without media change for the first 4 days. Media was completely changed after 6 days. Colonies were manually picked at day 10 to 96 wells. Clonal S6 cNESCs were maintained in 96 wells and passaged once 1:4 ratio for immunofluorescent labelling of cNESCs or in vitro differentiation assay. Cultures in 6F were fixed with 4% PFA solution for 10 min at room temperature.

**Immunocytochemistry.** Cultured cells were rinsed once with phosphate buffered saline and fixed with 4% PFA solution at room temperature for 15 min. For cell surface epitopes, the cells were stained without Triton X100 in PBS with 5 % foetal bovine serum. Intracellular epitopes were stained after 30 min permeabilisation with 0.3% Triton X100 in PBS with 5% foetal bovine serum and 1% BSA. Volocity demo v6.1.1 (Perkin Elmer) was used to quantify nuclear and cytoplasmic immunofluorescent stainings. Staining threshold was calculated from immunofluorescent staining of "epitope negative" cells or secondary antibody controls. Pictures were taken from minimum 6 randomly picked areas. Details of antibodies are listed in Supplementary Table 3.

**Western blot.** Cells for western blot experiments were replated the night before collection at sub confluent density, starved for 2 h for investigated cytokine, rinsed once with DMEM-F12 before treatment. Cells were lysed on ice with RIPA buffer containing protease and phosphatase inhibitors (Complete and PhosSTOP, Roche). Samples were centrifuged for 3 min at $15000 \times g$ to remove cell debris and treated with DNAse to degrade genomic DNA. Protein concentration was determined by BCA reaction (Thermo) according to the manufacturer's instructions. Equal amounts of protein (15–50 μg) were loaded in equal volumes to SDS-Polyacrylamide gradient gels. Proteins were transferred to nitrocellulose membranes and probed with indicated antibodies. The same membrane was stripped

and probed with various antibodies unless otherwise stated. Details of antibodies are listed in Supplementary Table 3.

**Animals used in the study.** We used for cNESC xenotransplantation experiments NSG mice (NOD.Cg-Prkdc-scid Il2rgtm1Wjl/SzJ (Strain #:005557 from The Jackson Laboratory), all animals were 8–10 weeks old females, total of 12 animals.

**Cell transplantation and immunohistochemistry.** In vivo differentiation of undifferentiated hNESCs cells was tested in NOD-SCID-Gamma adolescent mice. HNESCs cells (H1JA or CA1J) maintained in 6F media were collected with Accutase, centrifuged and resuspended in 6F media. After cell counting, cells were centrifuged and resuspended in DMEM-F12 with 10 μM Y27632 at $10^5$ cells per μl and kept on ice for maximum 2 h before injected into the wall of the lateral ventricle. A 26 gauge Hamilton syringe with a 45 degree bevelled tip was used to stereotaxically inject 1 μl of cell suspension at a rate of 0.1 μl/min at the coordinates: 0 AP, 1.4 ML, −1.7 DV, relative to bregma. The needle was left in place for 10 min after cell injection to prevent backflow and then slowly withdrawn. Mice were kept individually after surgery and monitored daily until sacrificed for tissue processing. Mice were anesthetized with avertin, perfused transcardially and postfixed for 2 h with 4% paraformaldehyde. Brains were cryopreserved in 30% sucrose, frozen and 18 μm horizontal cryostat sections were cut. Cryosections were blocked with 10% normal goat serum (NGS), 0.5% BSA and 0.3% triton and transplanted human cells were labelled with a primary antibody in PBS overnight at 4 °C, followed by an incubation with a secondary antibody and DAPI (5 μg/mL) in PBS for 1 h at RT. For double immunohistochemistry, sections were reblocked in 10% NGS, 0.5% BSA and 0.3% triton, incubated with a primary antibody 4 °C O/N and then with a secondary antibody at RT for 1 h. Details of antibodies are listed in Supplementary Table 3.

**In vitro electrophysiology.** Recording of action potentials and ionic currents of the iPS-derived neurons were carried out using standard whole-cell current- and voltage-clamp techniques (Richmond and Jorgensen, 1999). Cells were patched using fire-polished 3–7 MΩ resistant borosilicate pipettes (World Precision Instruments, USA) and membrane currents and voltages were recorded in the whole-cell configuration with a Digidata 1440 A and a MultiClamp 700 A amplifier, using the Clampex 10 software and processed with Clampfit 10 (Axon Instruments, Molecular Devices, USA). Data were digitized at 10–20 kHz and filtered at 2.6 kHz. To record spontaneous action potentials, the input current was held at 0 pA, and hyperpolarizing and depolarizing step or ramp currents were injected to elicit action potentials. To record voltage-gated K+ and Na+ currents, cells were held at −70 mV, and voltage steps from −90 mV to +60 mV were delivered at 10-mV increments. The pipette solution contained (in mM): K-gluconate 115; KCl 25; CaCl₂ 0.1; MgCl₂ 5; BAPTA 1; HEPES 10; Na₂ATP 5; Na₂GTP 0.5; cAMP 0.5; cGMP 0.5, pH7.2 with KOH, ~320 mOsm. The external solution consists of (in mM): NaCl 150; KCl 5; CaCl₂ 2; MgCl₂ 1; glucose 10; HEPES 10, pH 7.2–7.3 with NaOH, ~320–325 mOsm. Leak currents were not subtracted. All chemicals were from Sigma. Experiments were performed at RT (20–22 °C).

**Acute brain slices preparation.** Acute brain slices (200 μm thick) were prepared from the forebrain (coronally cut) from the transplanted NSG mice in ice-cold (~3 °C) oxygenated (95% O₂–5% CO₂) sucrose-based brain isolation solution (containing 120 sucrose, 26 NaHCO₃, 1 NaH₂PO₄, 2.5 KCl, 0.5 CaCl₂, 4 MgCl₂, 10 D-glucose, pH 7.4, 1 kynurenic acid, in mM) and kept at RT for a 1 h recovery period. After the recovery period the slices were transferred to artificial cerebrospinal fluid solution (aCSF) containing (in mM): 120 NaCl, 26 NaHCO₃, 1 NaH₂PO₄, 2.5 KCl, 2.5 CaCl₂, 2 MgCl₂, 10 D-glucose, pH 7.4. Kynurenic acid (1 mM) was added to block glutamate receptors, which might be activated during the dissection procedure.

**Ex vivo electrophysiology and data analysis.** For whole-cell patchclamp experiments, GFP-positive transplanted cells were selected in the mouse cortex. Slices were superfused at RT with HEPES-buffered external solution containing (in mM): 144 NaCl, 2.5 KCl, 10 HEPES, 1 NaH₂PO₄, 2.5 CaCl₂, 2 mM MgCl₂, 10 glucose, pH set to 7.4 with NaOH, continuously bubbled with 100% O₂. Cells were whole-cell clamped with electrodes containing a recording solution that comprised (in mM) of 130 K-gluconate, 4 NaCl, 0.5 CaCl₂, 10 HEPES, 10 BAPTA, 4 MgATP, 0.5 Na₂GTP, Alexa Fluor 594 pH set to 7.3 with KOH. Final osmolality was ~290 mOsm/kg. Recording electrodes had a resistance of 5–9 MΩ and the uncompensated series resistance was 40 ± 1 MΩ. Inclusion criteria was based on series resistance, leak current being lower than 500 pA and a stable baseline. Electrode junction potential (−14 mV) was compensated. A Multiclamp 700B (Molecular Devices) or Axopatch 200 (Molecular Devices) was used for voltage clamp data acquisition. Data were sampled at 50 kHz and filtered at 10 kHz using pClamp10.3 or 10.7 (Molecular Devices). Voltage-gated ion channels, series and membrane resistance, and membrane capacitance were analyzed using custom-written MATLAB scripts (MathWorks).

**Q-RT-PCR**. Total RNA was isolated from cultured cells with Qiagen RNeasy kit according to manufacturer's instructions. One ug of total RNA was reverse transcribed to cDNA with Qiagen reverse transcription kit. Quantitative RT-PCR detection of specific cDNA sequences was done with SYBR Green mix on a BIORAD thermocycler. Primer pairs for each cDNA target were tested for linear amplification and single product melting peak. Ct values are the average of 3 technical replicates, cDNA abundance is normalised to *GAPDH* housekeeping gene reads. Details of primer sequences are listed in Supplementary Table 3.

**Microarray analysis**. Total RNA was assessed for quality and quantity on a Bioanalyzer and global gene expression profiling performed with the Affymetrix microarray. Purified and labelled RNA was analyzed on Affymetrix Human Transcriptome Array 2.0 (Affymetrix) according to the manufacturer's instructions, at Microarray Facility of The Centre for Applied Genomics at Sickkids, Toronto, Ontario.

RMA method normalization, background substraction, and summarization was performed using Oligo (v1.34.2)[60] R package from bioconductor. Normalized gene expression intensities were log2-scaled for all subsequent analyses. Unless otherwise stated, all data presented are representative of at least two independent experiments. A linear model approach and the empirical Bayes statistics were implemented for differential gene expression analysis, as described in limma (v3.26.9)[61] R package user guide. P-values were adjusted using the Benjamini–Hochberg method and significance cut-off was set at 0.01. Euclidean distance matrix and complete linkage were used for hierarchical clustering of the samples using log2 transformed gene expression values. Euclidean distance matrix and Ward's minimum variance method of clustering were used to obtain the gene clusters using Z-score normalized gene expression values. All statistical analyses, Gene Ontology term and KEGG pathway analyses, and data visualization were done in R using R basic functions and the following packages: clusterProfiler (v2.4.3)[62], gplots (v3.0.1, https://CRAN.R-project.org/package=gplots), pathview (v1.10.1)[63] and stats (v3.2.2, http://www.rdocumentation.org/badges/version/stats). The microarray data generated in this study have been deposited in the GEO database under accession code GSE185258.

**Library preparation and long-read RNA-sequencing**. Cell lysates were homogenized with QIAshredder columns (QIAGEN #79654) and total RNA was extracted with RNeasy mini columns (QIAGEN #74104). RNA was DNase treated using TURBO DNase (ThermoFisher Scientific #AM2238). RNA quality was verified on a 2100 Bioanalyzer (Agilent Technologies) and polyadenylated RNA was enriched using NEBNext Poly(A) mRNA Magnetic Isolation Module (NEB #E7490L). Libraries were generated using 100 ng of poly(A) + RNA and Oxford Nanopore Technologies' Direct cDNA Sequencing Kit (ONT SQK-DCS109), with each sample barcoded with the Native Barcoding Expansion kits (ONT EXP-NBD104 and ONT EXP-NBD114), according to the manufacturer's specifications. Barcoded samples were pooled, loaded on a MinION sequencer on R9 Flow Cells (ONT FLO-MIN106D), and sequenced using the MinKNOW software v19. The long-read RNA-sequencing data generated in this study have been deposited in the GEO database under accession code GSE184081.

**Long-read RNA-seq data analysis**. Raw fast5 files were basecalled using Guppy v.4.2.2 with the guppy_basecaller command and the dna_r9.4.1_450bps_fast.cfg configuration file and default settings. Output fast5 files were generated with the -fast5_out argument. Barcodes were detected and reads were separated with the guppy_barcoder using the default settings. The resulting FASTQ reads were aligned to the GRCh38 human reference with Minimap2[64] using the following parameters: -aLx splice -cs=long. Raw read counts were obtained with the featureCounts tool from the Subread package v 2.0.0, using the exon counting mode[65].

EdgeR R-package (v3.12.1)[66] was then used to normalize the data, calculate RNA abundance (as counts per million reads (CPM)), and perform statistical analysis. Briefly, a common biological coefficient of variation (BCV) and dispersion (variance) were estimated based on a negative binomial distribution model. This estimated dispersion value was incorporated into the final EdgeR analysis for differential gene expression, and the generalized linear model (GLM) likelihood ratio test was used for statistics, as described in EdgeR user guide. Differential gene expression was established as genes with a fold change ≥ 1.5, p-value ≤0.05 and FDR ≤ 0.1.

**Single-cell RNA-sequencing analysis**. FastQ files obtained from cited publications[37,38] were aligned to the GRCh38 (hg38) human genome and GRCm38 (mm10) mouse genome where applicable, using the STAR aligner v2.7.5a[67] using the following parameters: -outStd SAM -outSAMunmapped Within -out-SAMstrandField intronMotif. Resulting BAM files were indexed and sorted with Samtools v1.11[68], and raw read counts were obtained for each single cell with the featureCounts tool as previously described. Cells with low read counts were excluded and counts in the remaining cells were then normalized to library size as counts per million reads (CPM) with the EdgeR R package. Genes with expression in less than 40 cells were excluded.

**Integration of single-cell RNA-seq and long-read RNA-seq datasets**. Long-read RNA-seq data for cNESCs (6F and 6F-2F) and for hESCs were first compared to highlight cNESC enriched genes. This gene list, in addition to known markers of cortical progenitors, was cross-referenced to the analyzed single-cell RNA-seq data[37,38] and visualized as heatmaps of single-cell and long-read RNA-seq expression (as Z-scores obtained from log2(CPM + 1)) using the Complex-Heatmaps R package[69]. Hierarchical clustering was performed using the Ward.D2 clustering method and Pearson clustering distance to separate single cell RNA-seq data into gene and cell clusters. Each gene cluster was then sorted according to expression in 6F cNESCs long-read RNA-seq data. For the Onorati et al. dataset[38], single cells were annotated with cell origin and age, while for the Yuzwa et al. dataset[37], single cells derived from the cortex were annotated (information available in the original publications). Single cells expressing the cortical progenitor markers SOX2, FOXG1, EMX2, PAX6 and LHX2 were also annotated. Cut-offs for considering a marker gene "expressed" in a cell were set by visualizing the gene expression distribution in each single-cell RNA-seq dataset (cut-off for Onorati et al. dataset: 10 CPM, cut-off for Yuzwa et al. datasets: 250 CPM). Distributions of gene expression across gene clusters was analyzed for 6F and 6F-2F cNESCs, using one-way ANOVAs followed by Tukey's multiple comparison test (p-value < 0.05 = *, < 0.01 = **, < 0.001 = ***, < 0.0001 = ****).

**Gene ontology term enrichment analyses**. GO term enrichment was assessed using the Database for Annotation, Visualization and Integrated Discovery (DAVID 6.8)[70]. Biological processes (BP) and pathway enrichment with Kyoto Encyclopedia of Genes and Genomes (KEGG) annotations were analyzed. A p-value < 0.05 and an FDR < 0.1 were set as cut-off for significant enrichment.

**Reporting summary**. Further information on research design is available in the Nature Research Reporting Summary linked to this article.

## Data availability

Data generated in this study were deposited to the Gene Expression Omnibus. Microarray and long-read RNA sequencing data are available under accession numbers GSE185258 and GSE184081, respectively. Publicly available datasets used in this study can be accessed under accession numbers GSE107122 and GSE81475. The KEGG[71,72] and DAVID[70] databases were used for gene ontology enrichment analysis. The data related to the manuscript are provided as Source Data files. Source data are provided with this paper.

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

## Acknowledgements

The authors thank Nagy laboratory members, Freda Miller, Tony Pawson, Jeffrey Wrana, Cindi Morshead, Michael Fehlings and Robert Hevner for antibodies, Maria Mileikovskaia for assistance with CA1 hESCs, Gordon Keller for H1, H7, H9 hESCs, Austin Smith for CB660 cell line, Ludovic Vallier for CTRL hiPSCs, Peter W Andrews for

SHEF6 hESCs, Chi-chung Hui for Q-PCR primers, Faustine Massin for DNA cloning, Chen He and Puzheng Zhang for technical assistance, Carla Mulas, Masaki Kinoshita, Ian Rogers, Natalie Payne and Kathryn Davidson for critical reading of the manuscript. This work was supported by grants from the Canadian Institutes of Health Research (CIHR) (CIHR - PJT- 378019) to S.M.I.H. S.M.I.H is a Junior 1 Research Scholar of the Fonds de Recherche du Québec - Santé (FRQ-S). G.K. is a recipient of an NSERC Postgraduate Scholarships – Doctoral (PGS D) and a Quebec Health Research Funds (FRQS) PhD training scholarship. Paul G. Allen Frontiers Group (Allen Distinguished Investigator program #12076; R.T.K, B.V.V.).

## Author contributions

B.V.V. designed and carried out the experiments, analysed and interpreted the data and wrote the manuscript, M.F. carried out transplantation experiments and analysed the data, H.Y. and E.L. carried-out experiments, S.G. did the in vitro electrophysiological recordings and interpreted the data, M.Z. interpreted the data, H.P. and R.T.K. carried out transplantation experiments and acute brain slice electrophysiological recordings and analysed the data, S.M.H. and G.K. analysed the microarray and long-read RNA-sequencing data, A.N. interpreted the data and wrote the manuscript.

## Competing interests

The authors declare no competing interests.
