## [Peer Review File · Nature Communications]

Signal requirement for cortical potential of transplantable human neuroepithelial stem cellsReviewers' Comments:

Reviewer #1:

Remarks to the Author:

The authors have established a novel combination of six factors that promote long-term proliferation of human neuroepithelial stem cells with cortical identity (cNECs) in vitro. Varga et al., initially present data on how the combined action of four previously described factors (FGF, BMPRI, TGFRI, GSK3i) is sufficient to induce cNECs from hESCs. By adding two more factors in order to inhibit β -catenin and AKT signalling, the authors establish a six-factor medium (6F) that induces high expression of forebrain markers like FOXP1 and OTX1/2, in addition to general neural stem cell markers like Sox2, Pax6 and Nestin. Moreover the 6F medium promotes the proliferation of cNECs for up to 15 passages. The combined addition of the factors to the medium regulates distinct signalling pathways with key roles in forebrain development. A transcriptome analysis of cNECs maintained in 6F medium showed that the profile of these cells is closer to samples with dorsal forebrain identity than midbrain identity or undifferentiated hPSCs. cNECs that are maintained under the suggested complement are capable to differentiate towards the three main cell lineages of the forebrain, neurons, astrocytes and oligodendrocytes following the physiological temporal order of cortical development. Moreover, the authors performed electrophysiology analysis and show that the cNEC-derived neurons exhibit action potential suggesting their functionality. Finally, it is provided evidence that cNECs are capable to proliferate and differentiate towards neurons following transplantation in mouse brain.

This is a well written manuscript that provides evidence on the signals supporting the generation of functional human neuroepithelial cells. The authors should address the following points:

Major points

The authors provide evidence that cNECs cultured with the 6F supplement for up to 15 passages maintain the expression of Foxg1 and that are capable to differentiate. Additional evaluation of a proliferative marker (Ki67 for example) would support the undifferentiated state of NECs in long term cultures.

Moreover, it is important that the authors perform an analysis for signs of senescence or genomic instability in long term cultures of cNECs with 6F.

A transcriptome analysis was performed to evaluate the gene expression profile of 6F-cNECs. A comparison with the transcriptional profile of human derived neuroepithelial or neural progenitor cells, rather than hPSCs differentiated towards forebrain neuroepithelium would strengthen the dorsal forebrain genetic signature of 6F cNECs.

A comparative transcriptomics analysis of the 6F and 4F NECs is also required as well as a comparison with the transcriptome of dorsal forebrain neuroepithelial cells from mouse.

The different Groups (1-4) of the cell lines that are written in the text (page 6) do not correspond to the same Group number in Figure 2E. The authors should correct accordingly.

While the authors provide sufficient data supporting that cNECs generate mature neurons upon differentiation, the question that is raised is whether they are able to form 3D organotypic cultures.

To investigate the potential of 6F cNECs to proliferate and differentiate in vivo the authors performed transplantation in mouse brain. Based on the expression of neural markers it is suggested that the transplanted cells are able to differentiate and express neuronal markers of the cortex. The authors should provide electrophysiological data for the neurons generated in vivo and derived from the 6F cNECs, to support their in vivo functionality.

The authors should provide higher magnification images of the positive cells expressing TBR1, CTIP1 and CUX1.

Minor points

Axis titles are confusing when referring to cells expressing specific markers. E.g. Foxg1+ve should be replaced by the simple Foxg1+.

In the figures they should indicate what is in the inserts. For example in Figure 2A it is not stated what

it is presented in the inserts. In general, the authors should present representative photos of their samples. In the panel of Figure 2C the inserts present the staining of DAPI, however this is not clear in the selected photos.

Figure 1 legend authors should clarify when 4F and 6F medium is used.

Figure 2 legend should include more details eg Fig. 2G

Reviewer #2:

Remarks to the Author:

The manuscript by Varga and colleagues addresses the proliferative capacity of the human pluripotent stem cells (hPSCs)-derived cortical-like neuroepithelial stem cells (NESCs) in vitro and the molecular pathways that drive the maintenance of their developmental potential. The authors found that activation of FGF and inhibition of both BMP and ACTIVIN A signaling are required for long-term NESC proliferation, and demonstrated that inhibition of GSK3, AKT, and nuclear CATENIN-b1 activity preserved dorsal telencephalon-specific potential. The manuscript presents generally convincing data and may aid the development of translational medicine by facilitating the development of stem cell-based therapeutic approaches to neurological disorders. As a manuscript answering fundamental questions that have potentially broad relevance to public health, this manuscript is likely to be an impactful contribution to the literature. However, the authors should address several concerns.

It is possible and perhaps likely that hPSC- and iPSC-derived NESCs, as well as the primary NESCs from human embryonic tissue, differ in terms of their transcriptome and biologic behavior. The authors should compare how their hPSCs-derived cortical-like NESCs compare when it comes to the cell culture media, differentiation potential and etc to the primary NESCs isolated from the human embryonic cerebral cortex and spinal cord by Onorati et al., 2016 (PMID: 27568284).

The authors should list the source and different lines of the human pluripotent stem cells in the main text, not just the Report Summary, and provide the information on which specific lines were used to generate specific data and figures.

The figures presented by the authors contain a great deal of information. I would ask that the authors carefully review their figure legends and data presentation to ensure everything is as simple, and described as well, as possible. For example, in Figure 1J, I am unclear what the red horizontal bar is. The authors should also add the relevant units used for the concentration of chemicals in Figure 1M, and I find no description of Figure 2A in the main text of the manuscript.

The authors should be clear and consistent with terminology. For example, supplement 1L is also called S1L, and in Supplemental Figure 1D I was unclear whether the data presented came from hESCs or hPSCs. In addition, the authors should take care to use consistent and up-to-date gene name nomenclature, for instance referring to CTIP2 and BRN2 by the official gene name, BCL11B and POU3F2, respectively.

I would ask that the authors confirm the absence of differential expression of NKX2.1 in hPSCs and D11 N1. Additionally, did the authors conduct statistical analysis for the genes listed, and, if not, why not?

To give a more representative impression on the existing literature, the authors should also consider the following, highly relevant references. For instance, Introduction paragraphs on page 2, the authors should consider adding the following references to the existing citations: Fuccillo et al., 2004 (PMID: 19560042); Molyneaux et al., 2007 (PMID: 18508260; PMID: 22492350); and Qian et al., 2000 (PMID: 23028117).

Statistical thresholds are not consistently reported. In some cases, significance is stated with $p <$

0.001. In others, $p < 0.01$ or 0.0001 . I appreciate the authors' efforts to provide this information, but on a quick glance at the figures, these differences are not apparent.

Authors response to referees' comments

Reviewer 1:

We thank the reviewer for the suggestions to extend the characterization of the cNECs in the 6F condition. We focused on two important properties of the cNECs; the transcriptional profile of cortical identity and the maturation of the transplanted cNEC derived neurons. We verified the cortical identity by marker expression, transcriptome comparison of *in vitro* mid-hindbrain NECs and *in vivo* rodent and human cortical progenitors. Please find the addressed comments below.

- 1. Additional evaluation of a proliferative marker (Ki67 for example) would support the undifferentiated state of NECs in long term cultures. Moreover, it is important that the authors perform an analysis for signs of senescence or genomic instability in long term cultures of cNECs with 6F.**

We thank the reviewer for this suggestion. We decided to label G1 arrested cells for P18/INK4A/CDKN1A and G1-G2 arrested cells for P21/CIP1/CDKN1A¹. We did not find any P18/INK4A/CDKN1A positive cells, and we could demonstrate that approx. 1-7% of the cells express P21/CIP1/CDKN1A at a high level in the culture (4 independent cNEC lines), indicating that the 6F condition keeps the vast majority of the cells cycling. We included the data in Supplementary Figure 2H,I.

- 2. A transcriptome analysis was performed to evaluate the gene expression profile of 6F-cNECs. A comparison with the transcriptional profile of human derived neuroepithelial or neural progenitor cells, rather than hPSCs differentiated towards forebrain neuroepithelium would strengthen the dorsal forebrain genetic signature of 6F cNECs.**

As suggested, we compared the transcriptome of cNECs to hiPS and human embryo-derived mid- hindbrain specified NES cells. It confirmed the dorsal forebrain specification of the cNECs and the lack of posterior gene expression that is characteristic of mhbNECs (Figure 5B). We had no access to live human embryonic telencephalic tissue from the proper age (gestational week 5-8). Therefore, we have chosen to compare our transcriptomic data to published datasets as also suggested by Reviewer #2. We have used single-cell RNA-sequencing data from mouse (Yuzwa et al., 2017)² and human (Onorati et al., 2016)³ embryonic cortices to identify the cNEC population in the tissue among other cells. We found the same cNECs that express FOXG1, SOX2, LHX2, EMX2 and PAX6 in the early developing embryonic telencephalon and using bioinformatic comparison of the two datasets. We could identify the common dorsal forebrain genes focusing mainly on transcription factors (FOXG1, EMX2, FEZF2, EMX1, SP8, PAX6, OTX2) with known role in the development and specification of the dorsal forebrain (Figure 5D,E). The detection of this region-specific gene expression and the differentiation potential of the cNECs verified that in the 6F condition, the cells are cortically specified and can self-renew.

- 3. A comparative transcriptomics analysis of the 6F and 4F NECs is also required as well as a comparison with the transcriptome of dorsal forebrain neuroepithelial cells from mouse.**

We detected changes in transcription factors that were shown *in vivo* to regulate the forebrain specification, such as OTX1/2, FOXG1 through the transcriptional regulation of multiple target genes. Therefore we wanted to detect the early changes of AKT and TNKS inhibitor removal to avoid the secondary effect of the changing transcriptome over multiple passages. We noticed the changes in FOXG1 protein and other transcription factor expression after four days. So, we collected mRNA samples after the 1st passage in the 4F or 6F-2F condition and compared their transcriptome to that of 6F cultured cells and cross-correlated with the published mouse and human embryonic datasets. We included the comparisons in Figure 5 and Supplementary Figures 5-8.

- 4. The different Groups (1-4) of the cell lines that are written in the text (page 6) do not correspond to the same Group number in Figure 2E. The authors should correct accordingly.**

We thank the reviewer for highlighting this matter. We changed the naming of the groups in Figure 1G to make it clear for the readers. Experimental groups on Figure 1G are identified with letters "a" to "d", and the groups on Figure 5A are identified with numbers 1 to 4.

- 5. While the authors provide sufficient data supporting that cNECs generate mature neurons upon differentiation, the question that is raised is whether they are able to form 3D organotypic cultures.**

We thank for the enthusiasm of the reviewer to test the potential of the cNECs to form organoids. In the current manuscript we have focused our experiments on demonstrating the self-renewal of single cNECs by colony formation capacity and verified differentiation potential using a monolayer culture. This reductionist system provided us with a better controlled *in vitro* model than a complex 3D organoid. We observed that 3D extracellular matrix products such as reduced growth factor containing Matrigel/Geltrex that is routinely used to make organoids from pluripotent stem cells, introduced undefined components from the Engelbreth-Holm-Swarm mouse sarcoma -either ECM and/or growth factors- that did not support a consistent and reproducible culture of cNECs among several lines. With the improvement and refining of organoid culture conditions, we will be able to test the capacity of cNECs to form cortical organoids in the future.

- 6. To investigate the potential of 6F cNECs to proliferate and differentiate in vivo the authors performed transplantation in mouse brain. Based on the expression of neural markers it is suggested that the transplanted cells are able to differentiate and express neuronal markers of the cortex. The authors should provide electrophysiological data for the neurons generated in vivo and derived from the 6F cNECs, to support their in vivo functionality.**

We thank the reviewer for this critical suggestion to prove the functional maturation of cNEC derived neurons *in vivo*. We knocked-in a CAG-EGFP cassette to the AAVS1 locus of the cNECs (H1JA and CA1J) that were used during the study and transplanted undifferentiated cNECs from 6F cultures to nod-scid gamma mouse cortices (8 weeks old) and used patch-clamp recording to verify the presence of voltage-gated sodium and potassium channels in the neuronal membrane, excitability to generate action potentials. Most importantly, we demonstrated that single neurons receive synaptic inputs from other neurons confirming the formation of functional synapses on the mature human neurons. We verified the human origin of the recorded neurons by co-labelling the Alexa 594 dye-filled cells with EGFP, STEM121 immunolabelling (please see Figure 6 K-O and verified MAP2 expression in human neurons in Figure 6K and Supplementary Figure 9H). We discuss in the revised manuscript that the transgenic labelling of human cells for long-term *in vivo* neuronal differentiation is challenging due to the silencing of transgenes introduced to cNECs by currently available methods. This makes the maturing live neuron identification in the xenografts very difficult. Still, we managed to identify cells that received synaptic inputs to proof the concept that *in vitro* cultured cNECs retain the ability to generate mature neurons *in vivo*.

- 7. The authors should provide higher magnification images of the positive cells expressing TBR1, CTIP1 and CUX1.**

We thank the reviewer for this suggestion. We modified the presented data to have high magnification images of cortical layer markers in the main figure (Figure 6 E-J) and added the low magnification images to Supplementary Figure 9A-F.

- 8. Axis titles are confusing when referring to cells expressing specific markers. E.g. Foxg1+ve should be replaced by the simple Foxg1+????**

We changed the labelling of the axis on the graphs to the suggested format.

In the figures they should indicate what is in the inserts. For example in Figure 2A it is not stated what is presented in the inserts. In general, the authors should present representative photos of their samples. In the panel of Figure 2C the inserts present the staining of DAPI, however this is not clear in the selected photos.????

We changed the figure panels with the DAPI or SOX2 inserts to provide a better format of the representative photos for the reader. Please find the new versions in Figure 3F,K; Supplementary Figure 2E

Figure 1 legend authors should clarify when 4F and 6F medium is used.????

We indicated in all figure legends the culture condition of the cells for each relevant panel.

Figure 2 legend should include more details eg Fig. 2G???? We removed 2G to give place for more relevant data.

In the revised manuscript we removed Figure 2G to give place for more relevant data.

Reviewer 2

We thank the reviewer for the suggestion to extend the transcriptomic characterization of the cNESCs in the 6F condition. Please find the addressed comments below.

- 1. It is possible and perhaps likely that hPSC- and iPSC-derived NESCs, as well as the primary NESCs from human embryonic tissue, differ in terms of their transcriptome and biologic behavior. The authors should compare how their hPSCs-derived cortical-like NESCs compare when it comes to the cell culture media, differentiation potential and etc to the primary NESCs isolated from the human embryonic cerebral cortex and spinal cord by Onorati et al., 2016 (PMID: 27568284).**

Thank you for the reviewer's suggestion to discuss the similarity of the 6F cultured cNES cells to that of Onorati et al, year. The study of Onorati aimed to identify neural cell types that are susceptible to Zika virus infection. Their use of human embryo derived neural progenitors from various ages takes advantage of the developmental regional specification of neural progenitors versus *in vitro* neural differentiation of pluripotent stem cells.

We used embryonic cNEsc single cell RNA-seq data from mouse (Yuzwa et al., 2017)² and human (Onorati et al., 2016)³ cortices and compared our *in vitro* cNEscs cultured in 6F and 4F media to these cells and verified the expression of dorsal forebrain transcription factors in the 6F cNEscs. We included our analysis in Figure 5C,D,E and Supplementary Figures 5-8). We present the clustering of the cultured NESCs from Onorati in Figure 5E, and very few cells express *PAX6*, *EMX2* and *LHX2* dorsal forebrain genes among the *FOXP1* positive cells, therefore we could not categorize them as cNEscs and we excluded the comparison of 6F cultures to these cells. Please find our reasons below.

Although Onorati concluded that they maintained cortical NES cells isolated from human embryonic dorsal forebrains from gestational weeks 5 and 6 they only succeeded with 2 from 3 isolations and provided data on the gene expression of early passage cells (passage 5-9). Our data support their observations that cNES cells in FGF and EGF can proliferate for up to 10 passages depending on the depending on the pluripotent stem cell line source, however, we detected reduced population doubling after passage 5 and variation in *SOX1* positive cell numbers (Figure 1 F, condition 3) and we also detected the reduction of *SOX2* positive cell ratio (60%) and the increased ratio of cells expressing high levels of GFAP, typical in radial glia like cells (data not shown). We confirmed the expression of dorsal forebrain markers (*FOXP1*, *LHX2*, *PAX6*, *OTX2*, *EMX2*) in all our cell lines (5) cultured in 6F media over multiple passages to ensure the conditions support the maintenance of dorsal forebrain specification (Figure 3F,G, 5B,D,E Supplementary Figure 2E,F). Importantly we tested the differentiation potential of the cNES cells maintained in 6F media and confirmed with all cell lines that they initiate the corticogenesis programme once we withdraw the six factors (Figure 3K,L, 4A-D, Supplementary Figure 2K,L, 3A-C). We show the sequential appearance of *TBR1*, *CTIP2*, *SATB2* cortical layer markers and GFAP glial marker mimicking the *in vivo* order of differentiation (Figure 4A,B,D, Supplementary Figure 3B,C), however, we could not find data for the cortex specific *TBR1* differentiation potential in Onorati et al. indicating that the culture condition cannot maintain the dorsal forebrain identity fully, since *CTIP2/BCL11b* is also expressed in the ventral forebrain and midbrain. In addition, we do not detect upregulation of ventral forebrain/ganglionic eminence specific genes like *GAD1* *GAD2* *DLX1* *DLX2* *DLX5* and detect mainly glutamatergic neurons in the 6F cultures; however the data presented by Onorati et al. shows the upregulation of ventral forebrain genes and downregulation of dorsal forebrain genes in the passage 9 NES cells further supporting the conclusion that FGF/EGF is not sufficient to maintain the cortical NES population in multiple genetic backgrounds. NES1 show outer radial glia-like genes expression signature (*FAM107A*, *GLI3*, *LIFR*, *HOPX*, *PTPRZ1*, *TNC*), NES2 show ventral forebrain (*GAD1* *GAD2* *DLX1* *DLX2* *DLX5*) and oRG signature (*FAM107A*, *LIFR*, *HOPX*, *PTPRZ1*, *TNC*) and glial precursor signature (GFAP^{high}, *GJA1*, *FABP7*, *ANXA1*, *S100B*, *SLC1A3*) and oligodendroglial precursor signature (*OLIG1*, *OLIG2*, *PLP1*) and NES3 reduced *PAX6* *SOX1*, *SOX2*.

- 2. The authors should list the source and different lines of the human pluripotent stem cells in the main text, not just the Report Summary, and provide the information on which specific lines were used to generate specific data and figures.**

We added all the information in the figure legends.

- 3. The figures presented by the authors contain a great deal of information. I would ask that the authors carefully review their figure legends and data presentation to ensure everything is as simple, and described as well, as possible. For example, in Figure 1J, I am unclear what the red horizontal bar is. The authors should also add the relevant units used for the concentration of chemicals in Figure 1M, and I find no description of Figure 2A in the main text of the manuscript.**

We apologise for the missing description of the red horizontal bar in Figure 1J and the concentration in Figure 1M, it has been corrected in the revised manuscript.

- 4. The authors should be clear and consistent with terminology. For example, supplement 1L is also called S1L, and in Supplemental Figure 1D I was unclear whether the data presented came from hESCs or hPSCs. In addition, the authors should take care to use consistent and up-to-date gene name nomenclature, for instance referring to CTIP2 and BRN2 by the official gene name, BCL11B and POU3F2, respectively.**

Thank you for the reviewer to point out the outdated gene names. We updated the the text and the figures with the latest gene names.

- 5. I would ask that the authors confirm the absence of differential expression of NKX2.1 in hPSCs and D11 N1. Additionally, did the authors conduct statistical analysis for the genes listed, and, if not, why not?**

We thank the reviewer for this suggestion. We could not find published ventral forebrain NES cells culture conditions with sustained ventral forebrain differentiation potential that would allow us to compare the expression levels of ventral and dorsal forebrain genes to that of cNESc genes. As a reference, we added the mRNA expression levels in the Supplementary Figure 1D for 2 neural differentiated hESC lines (H1 and H9) to indicate that the monolayer protocol does not induce high ventral forebrain or mid/ and hindbrain gene expression. We used three technical replicates for two cell lines. Therefore, we did not make statistical comparison. To verify that we do not have NKX2.1 immunopositive cells, we tried 2 antibodies that can label NKX2.1 cells in other tissues, but we did not detect any positive cells in 4 independent cNESc lines (H1JA, CA1J, S6 and LV); therefore, we did not show the absence of staining in the figure to save space for dorsal forebrain marker detections.

- 6. To give a more representative impression on the existing literature, the authors should also consider the following, highly relevant references. For instance, Introduction paragraphs on page 2, the authors should consider adding the following references to the existing citations: Fuccillo et al., 2004 (PMID: 19560042); Molyneaux et al., 2007 (PMID: 18508260; PMID: 22492350); and Qian et al., 2000 (PMID: 23028117).**

We thank the reviewer for suggesting missing relevant publications, we included them in the revised version of the manuscript.

- 7. Statistical thresholds are not consistently reported. In some cases, significance is stated with $p < 0.001$. In others, $p < 0.01$ or 0.0001 . I appreciate the authors' efforts to provide this information, but on a quick glance at the figures, these differences are not apparent.**

We thank the reviewer for raising this concern about the statistical values. We verified that all reported significance values are correct.

1. Satyanarayana, A. & Kaldis, P. Mammalian cell-cycle regulation: several Cdks, numerous cyclins and diverse compensatory mechanisms. *Oncogene* **28**, 2925-2939 (2009).
2. Yuzwa, S.A. et al. Developmental Emergence of Adult Neural Stem Cells as Revealed by Single-Cell Transcriptional Profiling. *Cell reports* **21**, 3970-3986 (2017).
3. Onorati, M. et al. Zika Virus Disrupts Phospho-TBK1 Localization and Mitosis in Human Neuroepithelial Stem Cells and Radial Glia. *Cell reports* **16**, 2576-2592 (2016).

Reviewers' Comments:

Reviewer #1:

Remarks to the Author:

The authors have addressed satisfactory the comments. They have characterised extensively the cell system of cNECs by providing additional immunofluorescent and transcriptomic data. Moreover, the authors performed a detailed analysis on the transplantation experiments, and they added functional analysis by recording action potentials. Minor comments have been also addressed and the new figures (microscopic photos and graphs) look great and easily followed by the reader.